# Clonal transcriptomics identifies mechanisms of chemoresistance and empowers rational design of combination therapies

**Sophia A Wild[†], Ian G Cannell[†], Ashley Nicholls, Katarzyna Kania, Dario Bressan, CRUK IMAXT Grand Challenge Team, Gregory J Hannon\*, Kirsty Sawicka\***

Cancer Research UK Cambridge Institute, University of Cambridge, Li Ka Shing Centre, Robinson Way, Cambridge, United Kingdom

**Abstract** Tumour heterogeneity is thought to be a major barrier to successful cancer treatment due to the presence of drug resistant clonal lineages. However, identifying the characteristics of such lineages that underpin resistance to therapy has remained challenging. Here, we utilise clonal transcriptomics with WILD-seq; **W**holistic **I**nterrogation of **L**ineage **D**ynamics by **seq**uencing, in mouse models of triple-negative breast cancer (TNBC) to understand response and resistance to therapy, including BET bromodomain inhibition and taxane-based chemotherapy. These analyses revealed oxidative stress protection by NRF2 as a major mechanism of taxane resistance and led to the discovery that our tumour models are collaterally sensitive to asparagine deprivation therapy using the clinical stage drug L-asparaginase after frontline treatment with docetaxel. In summary, clonal transcriptomics with WILD-seq identifies mechanisms of resistance to chemotherapy that are also operative in patients and pin points asparagine bioavailability as a druggable vulnerability of taxane-resistant lineages.

**\*For correspondence:**
kirsty.sawicka@cruk.cam.ac.uk (KS);
greg.hannon@cruk.cam.ac.uk (GJH)

[†]These authors contributed equally to this work

Group author details:
CRUK IMAXT Grand Challenge Team See page 26

## Editor's evaluation

This study advances a novel strategy of lineage tracing coupled with single-cell transcriptomics to allow unique insights into tumor heterogeneity and the diversity of response to treatment. These analyses reveal new insights into mechanisms of taxane resistance. Overall, the study is scientifically robust and puts forward a new methodology that will be of interest to scientists as well using this technology to gain insights into the factors that inform resistance to taxane treatment in an in vivo cancer model.

## Introduction

Intra-tumoural heterogeneity (ITH) is thought to underlie tumour progression and resistance to therapy by providing a reservoir of phenotypically diverse clonal lineages on which selective pressures from the microenvironment or therapeutic intervention exert their effects (*Bhang et al., 2015*; *Turajlic and Swanton, 2016*). Inference of clonal composition from bulk sequencing has elucidated the breadth of ITH across tumour types and suggests that often rare pre-existing clones can resist therapy-induced killing to drive relapse (*Dentro et al., 2021*; *Ding et al., 2012*; *Gerlinger et al., 2012*; *Jamal-Hanjani et al., 2014*; *Landau et al., 2013*). However, such methods are limited by their inability to characterise such resistant clones beyond genotype and how their properties change over time and in response to therapy. Recently, several lineage tracing approaches have emerged that are able to link clonal

**eLife digest** Cancer begins when a cell multiplies again and again to form a tumour. By the time that tumour measures a centimetre across, it can contain upwards of a hundred million cells. And even though they all came from the same ancestor, they are far from identical. The tumour's family tree has many branches, and each one responds differently to treatment. If some are susceptible to a drug the cells die, the tumour shrinks, and the therapy will appear to be successful. But, if even a small number of cancer cells survive, they will regrow, often more persistently, causing a relapse.

Identifying resistant cells, their characteristics, and how to kill them has been challenging due to a lack of good animal models. One way to keep track of a cancer family tree is to insert so-called genetic barcodes into the ancestral cells. As the tumour grows, the cells will pass the barcodes to their descendants. Scientists do this by using viruses that naturally paste their genes into the cells they infect. Applying this technique to an animal model of cancer could reveal which genes allow some cells to survive, and how to overcome them.

Wild, Cannell et al. developed a genetic barcoding system called WILD-seq and used it to track all the cells in a mouse tumour. The mice received the same drugs used to treat patients with breast cancer. By scanning the genetic barcodes using recently developed single cell sequencing technologies, Wild, Cannell et al. were able to identify and count each type of cancer cell and work out which genes they were using. This revealed which cells the standard treatment could not kill and exposed their genetic weaknesses. Wild, Cannell et al. used this information to target the cells with a drug currently used to treat leukaemia.

The drug identified by this new genetic barcoding approach is already licensed for use in humans. Further investigation could reveal whether it might help to shrink breast tumours that do not respond to standard therapy. Similar experiments could uncover more information about how other types of tumour evolve too.

identity with gene expression by utilising expressed genetic barcodes that are read-out by single cell RNA sequencing (*Biddy et al., 2018*; *Gutierrez et al., 2021*; *Quinn et al., 2021*; *Simeonov et al., 2021*; *Weinreb et al., 2020*; *Yang et al., 2022*). These powerful methods allow deconvolution of complex mixtures of clones while simultaneously providing a gene expression profile of those cells that can indicate the pathways on which they depend. However, to date in solid tumours these technologies have mostly been used to study drug response in vitro (*Gutierrez et al., 2021*; *Oren et al., 2021*) or metastatic dissemination in vivo (*Quinn et al., 2021*; *Simeonov et al., 2021*; *Yang et al., 2022*) and have not been utilised to study therapeutic response in immune-competent models.

A thorough understanding of the biomarkers of sensitivity and mechanisms of resistance to chemotherapy is essential if we are to improve patient outcomes. Most existing combination cancer therapies are not rationally designed but were instead empirically optimised to avoid overlapping toxicities. More recently alternative therapeutic strategies have emerged including synthetic lethality, drug synergy (*AlLazikani et al., 2012*; *ONeil et al., 2017*), and collateral sensitivity (*Mueller et al., 2021*; *Pluchino et al., 2012*; *Zhao et al., 2016*) that aim to leverage selective vulnerabilities of tumour cells while minimising toxicity. Of particular promise is collateral sensitivity, in which as a tumour becomes resistant to one drug it comes at the cost of sensitivity to a second drug. Since many modern clinical trials occur in the context of neo-adjuvant chemotherapy, the identification of frontline therapy-induced collateral sensitivities to second line therapy would have the potential to be rapidly translated into improved outcomes for patients.

Here, we describe WILD-seq (**W**holistic **I**nterrogation of **L**ineage **D**ynamics by **seq**uencing), an accessible and adaptable platform for lineage tracing at the single-cell transcriptomic level that facilitates in vivo analysis of clonal dynamics and apply it to the study of syngeneic triple negative breast cancer (TNBC) mouse models. Our optimised pipeline ensures recurrent representation of clonal lineages across animals and samples, facilitating analysis of clonal dynamics under the selective pressure of therapeutic intervention. Importantly, analysis of response of TNBC models to frontline taxane-based chemotherapy revealed an enrichment of clones with high levels of NRF2 signaling, implicating defense against oxidative damage as a major determinant of resistance to chemotherapy. Building on the work of others (*LeBoeuf et al., 2020*), we show that these NRF2-high, taxane-resistant

lineages are collaterally sensitive to asparagine deprivation as a result of L-asparaginase treatment and that they adapt to this second line intervention by up-regulating de novo asparagine synthesis by increasing asparagine synthetase (*Asns*) expression. Together these data indicate that high levels of NRF2 signaling, which is also observed in patients following neo-adjuvant chemotherapy, promotes both resistance to chemotherapy and sensitivity to asparagine deprivation and warrant the exploration of L-asparaginase as a therapeutic modality in solid tumours.

## Results

### Establishment of an expressed barcode system to simultaneously detect clonal lineage and gene expression

WILD-seq uses a lentiviral library to label cells with an expressed, heritable barcode that enables identification of clonal lineage in conjunction with single cell RNA sequencing. The WILD-seq construct comprises a zsGreen transcript which harbours in its 3′ untranslated region (UTR) a barcode consisting of two 12 nucleotide variable regions separated by a constant linker (*Figure 1a*). Each variable region is separated from any other sequence in the library by a Hamming distance of 5 to allow for library preparation and sequencing error correction and we detected over 2.7 million unique barcodes in our vector library by sequencing after clustering based on Hamming distance. The barcode is appropriately positioned relative to the polyadenylation signal to ensure its capture and sequencing by standard oligo-dT single-cell sequencing platforms.

The standard WILD-seq pipeline is illustrated in *Figure 1b*. A heterogeneous cell line is transduced with a barcode library at low multiplicity of infection (MOI) to ensure that each cell receives a maximum of one barcode. An appropriate size pool of barcoded clones is selected and stabilised in culture. Empirically, we have found that three separate pools each established from 250 individual clones, that are maintained separately and combined immediately prior to implantation, works well to provide effective representation of the diversity within the cell lines used herein, while also enabling recurrent representation of the same clones across animals and experiments. Once stabilised in culture, the pool of WILD-seq clones can be analysed directly by single-cell sequencing or injected into a recipient animal for in vivo tumour growth. WILD-seq single-cell sequencing libraries can be prepared using a standard oligo-dT-based protocol and addition of an extra PCR amplification step can be used to increase coverage of the barcode region and aid cell lineage assignment.

We first established a WILD-seq clonal pool from the mouse 4T1 cell line, a triple negative mammary carcinoma model that can be orthotopically implanted into the mammary fat pad of a BALB/c syngeneic host, which we have previously shown to be heterogeneous with distinct sub-clones having unique biological properties (*Wagenblast et al., 2015*). We performed single-cell sequencing of the in vitro WILD-seq pool (*Figure 1c*) and in vivo tumours derived from this clonal pool (*Figure 1d*). Over the course of our studies, we injected multiple cohorts of mice with our WILD-seq 4T1 pool as detailed in *Supplementary file 1*, some of which were subjected to a specific drug regime. All tumours were harvested at humane endpoint, as determined by tumour volume unless otherwise stated and immediately dissociated for single-cell sequencing.

For the purpose of characterising the baseline properties of our clones, we performed an in-depth transcriptomic analysis of all tumours from vehicle-treated animals. A WILD-seq barcode and thereby clonal lineage could be unambiguously assigned to 30–60% of cells per sample within the presumptive tumour cell/mammary epithelial cell cluster. A total of 132 different WILD-seq barcodes were observed in vitro and in total 94 different WILD-seq barcodes were observed across our in vivo tumour samples. Our in vivo tumour samples comprised both tumour cells and host cells of the tumour microenvironment including cells of the innate and adaptive immune system, enabling simultaneous profiling of the tumour and its microenvironment (*Figure 1d*). Clustering was performed after removal of reads mapping to the WILD-seq vector, to avoid any influence of the WILD-seq transcript on clustering, and the WILD-seq barcode assignment subsequently overlaid onto these data. The tumour cell clusters were clearly identifiable by the high expression of the barcode transcript (*Figure 1d*). Occasionally a barcode was observed in cells which clustered according to their transcriptome outside of the main tumour cluster. Since this could be the result of sequencing or technical error causing a mismatch between the WILD-seq barcode and the cell of origin, only barcoded cells that clustered within the main tumour/mammary epithelium cell cluster were included in our analysis.

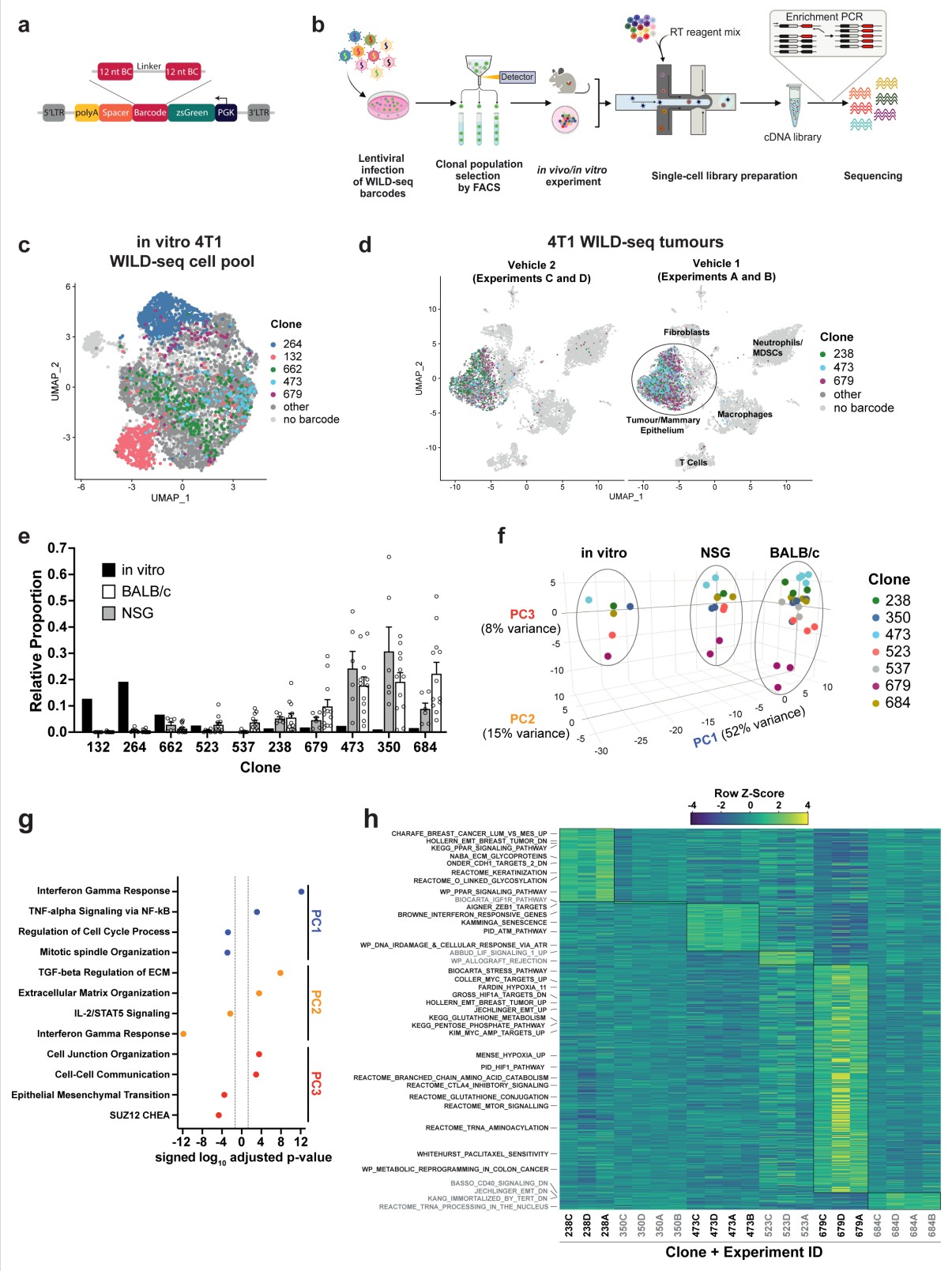

**Figure 1.** Establishment of an expressed barcode system to simultaneously detect clonal lineage and gene expression from single cells in vivo. (**a**) Lentiviral construct design. An attenuated PGK promoter drives expression of a transcript encoding zsGreen and harboring a WILD-seq barcode sequence in the 3′ UTR. A spacer sequence and polyadenylation signal ensure that that the barcode is detectable as part of a standard oligo dT single-cell RNA library preparation and sequencing pipeline. The barcode cassette comprises 2 distinct 12 nucleotide barcode sequences separated by a

*Figure 1 continued on next page*

*Figure 1 continued*

constant 20 nucleotide linker region. The library of barcode sequences was designed with Hamming distance 5 to allow for sequencing error correction. (**b**) Schematic of WILD-seq method. Tumour cells are infected with the WILD-seq lentiviral library and an appropriate size population of zsGreen positive cells isolated, each of which will express a single unique WILD-seq barcode. This WILD-seq barcoded, heterogenous cell pool is then subjected to an intervention of interest (such as in vivo treatment of the implanted pool with a therapeutic agent) and subsequently analysed by single cell RNA sequencing using the 10x Genomics platform. An additional PCR amplification step is included that specifically enriches for the barcode sequence to increase the number of cells to which a WILD-seq barcode can be conclusively assigned. (**c**) scRNA-seq of in vitro 4T1 WILD-seq cell pool. UMAP plot of in vitro cultured 4T1 WILD-seq cells. Cells for which a WILD-seq clonal barcode is identified are shown as dark grey or coloured spots. Cells which belong to five selected clonal lineages are highlighted. (**d**) scRNA-seq of 4T1 WILD-seq tumours. UMAP plots of vehicle-treated 4T1 WILD-seq tumours generated by injecting the 4T1 WILD-seq pool into the mammary fatpad of BALB/c mice. Four independent experiments were performed each involving injection into three separate host animals. Six animals from experiments A and B received vehicle 1 (10% DMSO, 0.9% β-cyclodextrin) and six animals from experiments C and D received vehicle 2 (12.5% ethanol, 12.5% Kolliphor). (**e**) Clonal representation. Proportion of tumour cells assigned to each clonal lineage based on the WILD-seq barcode (n=1 for in vitro cultured cells, n=6 for tumours from NSG mice, n=12 for vehicle-treated tumours from BALB/c mice). Selected clones from the most abundant lineages are plotted. Data represents mean ± SEM. (**f**) Principal component analysis of clonal transcriptomes. Pseudo-bulk analysis was performed by summing counts for all tumour cells expressing the same WILD-seq clonal barcode within an independent experiment. For in vivo tumour samples each point represents the combined cells from three animals. Principal component analysis of normalized pseudo-bulk count data showed separation of samples by origin with PC1 and PC2 and separation by clonality with PC3. (**g**) Transcriptomic programs associated with principal components. The top/bottom 50 gene loadings of PC1, PC2, and PC3 were analyzed using Enrichr (*Chen et al., 2013*; *Kuleshov et al., 2016*; *Xie et al., 2021*). (**h**) Clonal transcriptomic signatures from vehicle-treated BALB/c tumours. An AUCell score (*Aibar et al., 2017*) enrichment was calculated for each clone and for each experiment by comparing cells of a specific clonal lineage of interest to all assigned tumour cells within the same experiment. All gene sets which showed consistent and statistically significant enrichment in one of the six most abundant clones across experiments are illustrated.

We reproducibly observed the same clonal populations across animals and independent experiments which is critical to our ability to examine the effects of different interventions and treatments (*Figure 1d and e*). The relative abundance of clones was similar in tumours grown in NOD scid gamma (NSG) immunodeficient and BALB/c immunocompetent mice but was drastically different to that found in the in vitro cell pool from which they were established (*Figure 1e*, *Supplementary file 2*), suggesting that clones that show greatest fitness in cell culture do not necessarily show fitness in vivo. Therefore, in vitro clonal lineage tracking experiments are likely to capture a different collection of clones and have the potential to identify sensitive or resistance clones that are not represented in vivo. Pseudo-bulk analysis of the major clonal lineages revealed that the composition of the tumour microenvironment has a dramatic effect on the transcriptome of the tumour cells for all clones (*Figure 1f*). Comparison of in vitro culture, tumours from NSG mice, and tumours from BALB/c mice by principal component analysis (PCA), showed clear separation of the tumour cells depending on their environment, with differences in interferon gamma signaling, TNF-alpha signaling, and cell cycle being most prominent between cells grown in vivo and in vitro (PC1, *Figure 1g*). Differences in gene expression between tumours growing in immunocompetent and immunodeficient hosts were related to changes in the expression of extracellular matrix proteins and changes in interferon gamma and Il-2 signaling, consistent with the differences in T-cell abundance (PC2, *Figure 1g*). These data highlight the importance of the host immune system in sculpting the transcriptome and provide cautionary context for the analysis of tumour gene expression in immune-compromised hosts. Although there were large differences between clonal gene expression patterns across hosts, the clones showed consistent differences in gene expression across all settings, reflective of intrinsic clonal properties, with the biggest variation in gene expression across the clones being related to their position along the epithelial-mesenchymal transition (EMT) axis (PC3, *Figure 1g*). In particular, Clone 679 is the most distinct and the most mesenchymal of the clones.

To further characterise the major clones in our tumours, we performed gene set expression analysis using AUCell (*Aibar et al., 2017*) to identify pathways that are enriched in cells of a specific clonal lineage. Analysis was performed across four independent experiments each with three vehicle-treated animals and for the majority of clones we were able to identify distinct gene expression signatures that were reproducible across animals and experiments (*Figure 1h*, *Supplementary file 4*, *Supplementary file 5*).

# Simultaneous detection of changes in clonal abundance, gene expression, and tumour microenvironment in response to BET bromodomain inhibition with WILD-seq

Having established that we can repeatedly observe the same clonal lineages and their gene expression programs across animals and experiments, we next sought to perturb the system. We chose the BET bromodomain inhibitor JQ1 for our proof-of-principle experiments to assess the ability of the WILD-seq system to simultaneously measure changes in clonal abundance, gene expression and the tumour microenvironment that occur following therapeutic intervention. JQ1 competitively binds to acetylated lysines, displacing BRD4 and thereby repressing transcription at specific loci. A large number of studies have indicated that BET inhibitors may be beneficial in the treatment of hematological malignancies and solid tumours including breast cancer, possibly by inhibiting certain key proto-oncogenes such as MYC (*Jiang et al., 2020*).

Treatment of our 4T1 WILD-seq tumour-bearing mice with JQ1 caused an initial suppression of tumour growth but with only a small overall effect on time to humane endpoint (*Figure 2a*). Tumours treated with JQ1 or vehicle alone were harvested at endpoint, dissociated and subjected to single-cell sequencing (*Figure 2b*). Two independent experiments were performed, each with 3 mice per condition.

We first explored whether JQ1 had any effect on the tumour microenvironment. The most striking difference we observed was a change in abundance among the cells belonging to the T-cell compartment. To analyze this further, we computationally extracted these cells from the single cell data, reclustered them and performed differential abundance testing using Milo (*Figure 2c*). Milo detects sets of cells that are differentially abundant between conditions by modeling counts of cells in neighborhoods of a KNN graph (*Dann et al., 2022*). When applied to our reclustered T-cells, Milo identified a significant decrease in abundance in cytotoxic T-cells, as identified by their expression of *Cd8a* and *Cd8b1*, following JQ1 treatment. A significant change was observed in both of our experiments although the magnitude of the effect was greater in experiment A (*Figure 2c*).

We next examined the effect of JQ1 treatment on the transcriptome of the tumour cells. Differential expression analysis was performed for each clonal lineage and experiment independently. As expected, given its mode of action, we identified significant down-regulation of a wide range of genes with consistent changes across clonal lineages (*Figure 2d*, *Supplementary file 6*). Among the repressed genes, were a number of genes related to interferon (IFN) signaling and antigen processing and presentation (*Figure 2d and e*), including *Gbp2* which is strongly induced by IFN gamma, the MHC class II protein, *Cd74*, and *B2m*, a component of the MHC class I complex. JQ1 has previously been reported to directly inhibit transcription of IFN-response genes (*Gibbons et al., 2019*; *Gusyatiner et al., 2021*) suggesting this may be due to the direct action of JQ1 within our tumour cells; however, JQ1-dependent changes to the tumour microenvironment may also influence these expression pathways.

Our barcoded 4T1 clones showed varied sensitivity to JQ1, with treatment causing reproducible changes to clonal proportions within the tumour (*Figure 2f and g*, *Figure 2—figure supplement 1a* and *Supplementary file 2*). *Figure 2f* shows the proportions of clones in vehicle or JQ1 treated tumours from a representative experiment. We classified one of the most abundant clones, clone 473, as highly sensitive to JQ1 treatment and three clones as being resistant to JQ1 treatment, clones 93, 439, and 264 based on these clones showing consistent behavior across two independent experiments (see *Figure 2—figure supplement 1a* for a side-by-side comparison of experiments). These resistant clones which together make up less than 5% of the tumour in vehicle treated mice constitute on average 12.8% of the JQ1-treated tumours. To correlate the baseline transcriptomic signatures of clones with JQ1-sensitivity and resistance, we derived a $\log_2$ fold change value for each clone using data from two independent experiments, each consisting of 3 vehicle and 3 JQ1 treated animals (*Figure 2g*). We used these fold change values as a measure of JQ1 response to investigate transcriptomic characteristics that are correlated with sensitivity and resistance, an approach that obviates the need for binary classification of clones as sensitive and resistant and takes into account all the available data. We identified gene sets whose expression in vehicle-treated tumours was highly correlated with response (*Figure 2h and i*, *Supplementary file 7*). Interestingly, interferon signaling which is significantly attenuated in our JQ1-treated tumours is highly correlated with sensitivity to JQ1, suggesting a possible higher dependence of the sensitive clones on these pathways. Conversely

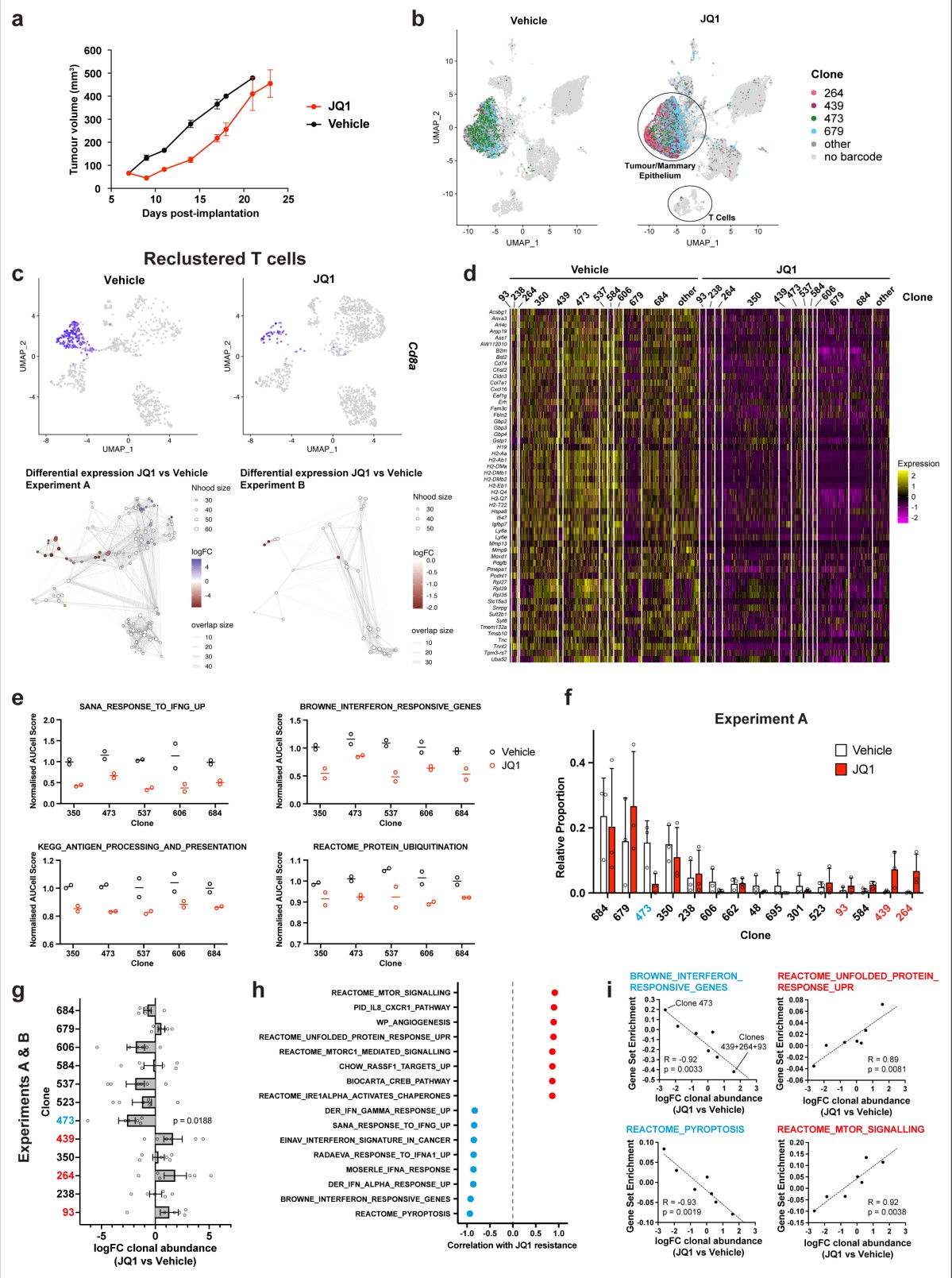

**Figure 2.** Simultaneous detection of changes in clonal abundance, gene expression, and tumour microenvironment in response to BET bromodomain inhibition with WILD-seq. (**a**) Tumour growth curves with JQ1 treatment. 4T1 WILD-seq tumours were treated with the BET bromodomain inhibitor JQ1 or vehicle from 7 days post-implantation until endpoint (n=4 mice per condition). 75 mg/kg JQ1 (dissolved in DMSO and diluted 1:10 in 10% β-cyclodextrin) 5 days/week (5 consecutive days followed by 2 days off). Data represents mean ± SEM. (**b**) scRNA-seq of JQ1-treated 4T1 WILD-seq

*Figure 2 continued*

tumours. UMAP plots of vehicle- or JQ1-treated 4T1 WILD-seq tumours. Combined cells from 2 independent experiments, each with 3 mice per condition are shown. Cells for which a WILD-seq clonal barcode is identified are shown as dark grey or coloured spots. Cells which belong to four selected clonal lineages are highlighted. (**c**) JQ1 treatment results in a reduction in Cd8+ T cells within 4T1 tumours. Cells belonging to the T-cell compartment were computationally extracted from the single cell data and reclustered. Upper panels show combined UMAP plots from experiments A and B with Cd8a expression per cell illustrated enabling identification of the Cd8+ T cell cluster. Lower panels show neighbourhood graphs of the results from differential abundance testing using Milo (***Dann et al., 2022***). Coloured nodes represent neighbourhoods with significantly different cell numbers between conditions (FDR <0.05) and the layout of nodes is determined by the position of the neighbourhood index cell in the UMAP panel above. Experiments A and B were analysed separately due to differences in cell numbers. (**d**) Differential gene expression between JQ1- and vehicle-treated tumour cells. Single cell heatmap of expression for genes which are significantly and consistently down-regulated across clonal lineages (combined fisher p-value <0.05 and mean logFC <–0.2 for both experiments).1600 cells are represented (400 per experiment/condition), grouped according to their clonal lineage. (**e**) Differential gene set expression between JQ1 and vehicle-treated tumour cells. Median AUCell score per experiment/condition for selected gene sets. The five clonal lineages with the highest representation across experiments are shown. (**f**) Clonal representation. Proportion of tumour cells assigned to each clonal lineage in experiment A based on the WILD-seq barcode (n=3 tumours per condition). Clones which make up at least 2% of the assigned tumour cells under at least one condition are plotted. The most sensitive clone 473 is highlighted in blue and the most resistant clones 93, 439, 264 are highlighted in red. Data represents mean ± SD. (**g**) Clonal response to JQ1-treatment. Log$_2$ fold change in clonal proportions upon JQ1 treatment across experiments A and B. Fold change was calculated by comparing each JQ1-treated sample with the mean of the three corresponding vehicle-treated samples from the same experiment. p-value calculated by one-sample t-test vs a theoretical mean of 0. Data represents mean ± SEM. (**h and i**) Correlation of JQ1-response with baseline clonal transcriptomic signatures. Clonal gene set enrichment scores for vehicle-treated tumours were calculated by comparing cells of a specific clonal lineage of interest to all assigned tumour cells within the same experiment. Correlation between these scores and JQ1-treatment response (mean log$_2$ fold change clonal proportion JQ1 vs vehicle) was then calculated for each gene set. Selected gene sets with the highest positive or negative correlation values (Pearson correlation test) are shown. A positive correlation indicates a higher expression in resistant clones, whereas a negative correlation indicates a higher expression in sensitive clones. Resistant clonal lineages identified by barcodes 93, 264, and 439 were combined for the purpose of this analysis to have enough cells for analysis within the vehicle-treated samples.

The online version of this article includes the following figure supplement(s) for figure 2:

**Figure supplement 1.** Clonal representation (related to *Figure 2g*) and correlation of individual UPR pathways with JQ1 response (related to *Figure 2i*).

resistance is associated with higher levels of unfolded protein response (UPR), in particular the IRE1 branch of the UPR (***Figure 2h and i*** and ***Figure 2—figure supplement 1b***), and mTOR signaling consistent with a known role of mTOR-mediated autophagy in resistance to JQ1 (***Luan et al., 2019***), and cytotoxic synergy between PI3K/mTOR inhibitors and BET inhibitors (***Lee et al., 2015***; ***Stratiko-poulos et al., 2015***).

## Clonal transcriptomic correlates of response and resistance to taxane chemotherapy in the 4T1 mammary carcinoma model

Our studies with JQ1 exemplify the ability of the WILD-seq system to simultaneously measure in vivo the effect of therapeutic intervention on clonal dynamics, gene expression and the tumour micro-environment. However, we were interested in using our system to investigate a chemotherapeutic agent currently in use in the clinic. We therefore treated our 4T1 WILD-seq tumour-bearing mice with docetaxel as a representative taxane, a class of drugs which are routinely used to treat triple negative breast cancer patients. As with JQ1, docetaxel treatment resulted in an initial, modest reduction in tumour growth followed by relapse (***Figure 3a***). Comparison of vehicle and docetaxel (DTX) treated tumours revealed differential response of clonal lineages to treatment (***Figure 3b, c and d***, ***Figure 3—figure supplement 1*** and ***Supplementary file 2***). ***Figure 3c*** shows the proportions of clones in vehicle or docetaxel treated tumours from a representative experiment. We classified clone 679 as docetaxel resistant and clone 238 as docetaxel sensitive based on these clones showing consistent behaviour across independent experiments (see ***Figure 3—figure supplement 1a*** for side-by-side comparison of experiments).

To correlate the clones' baseline transcriptomic profiles with response to docetaxel, we derived a log$_2$ fold change value for each clone, from two independent experiments each consisting of three vehicle and three docetaxel-treated animals (***Figure 3d***), as we did for JQ1. This approach revealed a major role for EMT in modulating sensitivity and resistance to taxane-based therapy. The 4T1 clones which are most sensitive to docetaxel are characterised by high expression of E-cadherin regulated genes and low ZEB1 activity consistent with a more epithelial phenotype (***Figure 3e and f***, ***Supplementary file 8***). These observations are in agreement with previous studies that have implicated EMT,

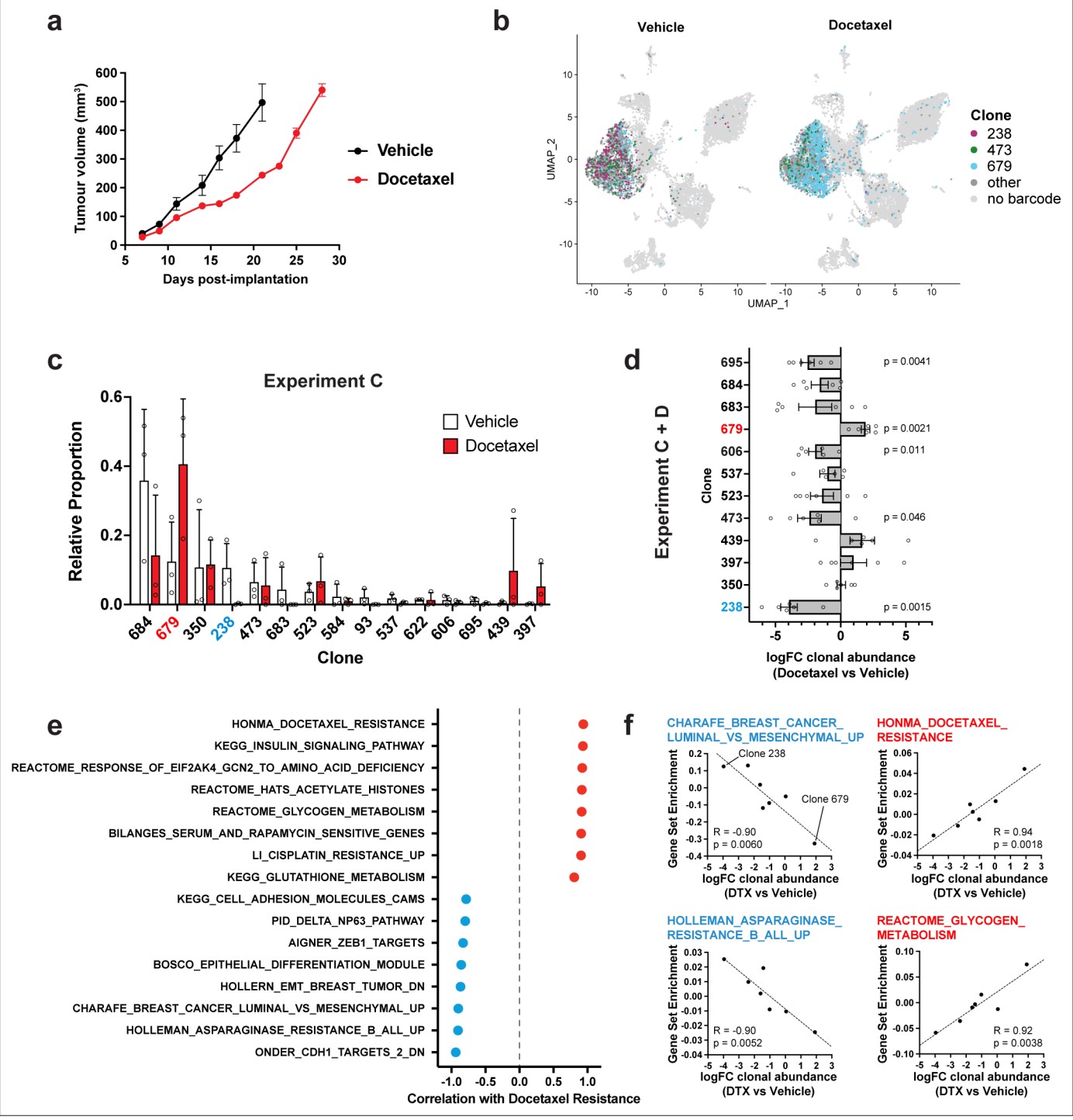

**Figure 3.** Clonal transcriptomic correlates of response and resistance to taxane chemotherapy in the 4T1 mammary carcinoma model. (**a**) Tumour growth curves with docetaxel treatment. 4T1 WILD-seq tumours were treated with docetaxel or vehicle (12.5% ethanol, 12.5% Kolliphor) from 7 days post-implantation for 2 weeks (n=5 mice per condition). Dosing regimen was 12.5 mg/kg docetaxel three times per week. Data represents mean ± SEM. (**b**) scRNA-seq of docetaxel-treated 4T1 WILD-seq tumours. UMAP plots of vehicle- or docetaxel-treated 4T1 WILD-seq tumours. Combined cells from 2 independent experiments, each with 3 mice per condition are shown. Cells for which a WILD-seq clonal barcode is identified are shown as dark grey or coloured spots. Cells which belong to three selected clonal lineages are highlighted. (**c**) Clonal representation. Proportion of tumour cells assigned to each clonal lineage in experiment C based on the WILD-seq barcode (n=3 tumours per condition). Clones which make up at least 2% of the assigned tumour cells under at least one condition are plotted. The most sensitive clone 238 is highlighted in blue and the most resistant clone 679 is highlighted in red. Data represents mean ± SD. (**d**) Clonal response to docetaxel-treatment. Log₂ fold change in clonal proportions upon docetaxel treatment across

*Figure 3 continued on next page*

*Figure 3 continued*

experiments C and D. Fold change was calculated by comparing each docetaxel-treated sample with the mean of the three corresponding vehicle-treated samples from the same experiment. p-values calculated by one-sample t-test vs a theoretical mean of 0. Data represents mean ± SEM. (**e and f**) Correlation of docetaxel-response with baseline clonal transcriptomic signatures. Clonal gene set enrichment scores for vehicle-treated tumours were calculated by comparing cells of a specific clonal lineage of interest to all assigned tumour cells within the same experiment. Correlation between these scores and docetaxel-treatment response (mean log$_2$ fold change clonal proportion docetaxel vs vehicle) was then calculated for each gene set. Selected gene sets with the highest positive or negative correlation values (Pearson correlation test) are shown. A positive correlation indicates a higher expression in resistant clones, whereas a negative correlation indicates a higher expression in sensitive clones.

The online version of this article includes the following figure supplement(s) for figure 3:

**Figure supplement 1.** Clonal representation (related to *Figure 3d*).

and its associated endowment of cancer stem cell-like characteristics, as a mechanism of resistance to cytotoxic chemotherapies like docetaxel in cell culture and patients (*Bhola et al., 2013*; *Creighton et al., 2009*; *Gupta et al., 2009*). Resistance to docetaxel was highly correlated with up-regulation of multiple gene sets (*Figure 3e and f*, *Supplementary file 8*). This included genes whose expression is elevated in non-responders to docetaxel in human breast cancer patients (*Honma et al., 2008*) demonstrating the relevance of findings arising from this approach. Interestingly, we also identify metabolic reprogramming as a potential mechanism of docetaxel resistance with higher expression of genes related to glycogen and glutathione metabolism being correlated with resistance to docetaxel (*Figure 3e*).

## Clonal transcriptomic signatures of response and resistance to taxane chemotherapy in the D2A1-m2 mammary carcinoma model

To explore the general applicability of WILD-seq to other models, we utilized a second triple negative mammary carcinoma model, D2A1-m2. Similar to the 4T1 cell line, this line was derived from a mouse mammary tumour in a BALB/c mouse and can be orthotopically implanted into the mammary fat pad of immunocompetent, syngeneic hosts (*Jungwirth et al., 2018*).

We established a WILD-seq D2A1-m2 clonal pool by transducing the D2A1-m2 cell line with our WILD-seq barcode library. These barcoded cells were orthotopically implanted into a cohort of mice, half of which were treated with docetaxel, while the remaining animals received vehicle only. The response to docetaxel treatment was comparable to that observed for 4T1 tumours, with an initial reduction in tumour growth with subsequent relapse (*Figure 4a*). We performed single-cell RNA sequencing of three tumours per condition and assigned the tumour cells to a distinct clonal lineage based on the presence of the WILD-seq barcode (*Figure 4b*). In total 103 different WILD-seq barcodes were observed in vivo with a dramatic shift in relative clonal abundance on docetaxel treatment (*Figure 4b and d*, *Supplementary file 3*). Unlike our 4T1 breast cancer model, variation between clonal lineages was no longer dominated by the EMT status of the clones and all clones exhibited a more mesenchymal-like phenotype consistent with the fact that this was a subline of D2A1 selected for its metastatic properties (*Figure 4c*). This provides us with a distinct yet complementary system to investigate chemotherapy resistance with the potential to reveal alternative mechanisms than EMT status.

We identified three clones which were acutely sensitive to docetaxel, clones 118, 2874, and 1072. Together these constitute on average 37% of the vehicle-treated tumours but only 1.3% of the docetaxel-treated tumours (*Figure 4d*). To understand the properties of these clones, we analysed the baseline gene expression characteristics of clones in vehicle-treated tumours. The gene expression of cells from a clone of interest was compared to all tumour cells to which a WILD-seq barcode could be assigned from the same sample, and clonal signatures identified that were significantly enriched across animals. Specific gene expression signatures were identifiable for all clones analysed, some of which were unique to a single clone while others overlapped across the sensitive clones (*Figure 4e*, *Supplementary file 9*, *Supplementary file 10*). For example, clone 1072 shows elevated levels of expression of cell cycle related pathways, such as E2F-target genes (*Figure 4f*), indicating that aberrant cell cycle control in these cells could increase their susceptibility to an antimitotic cancer drug, interestingly high levels of E2F-targets have recently been shown to be associated with response to chemotherapy in breast cancer patients (*Sammut et al., 2022*).

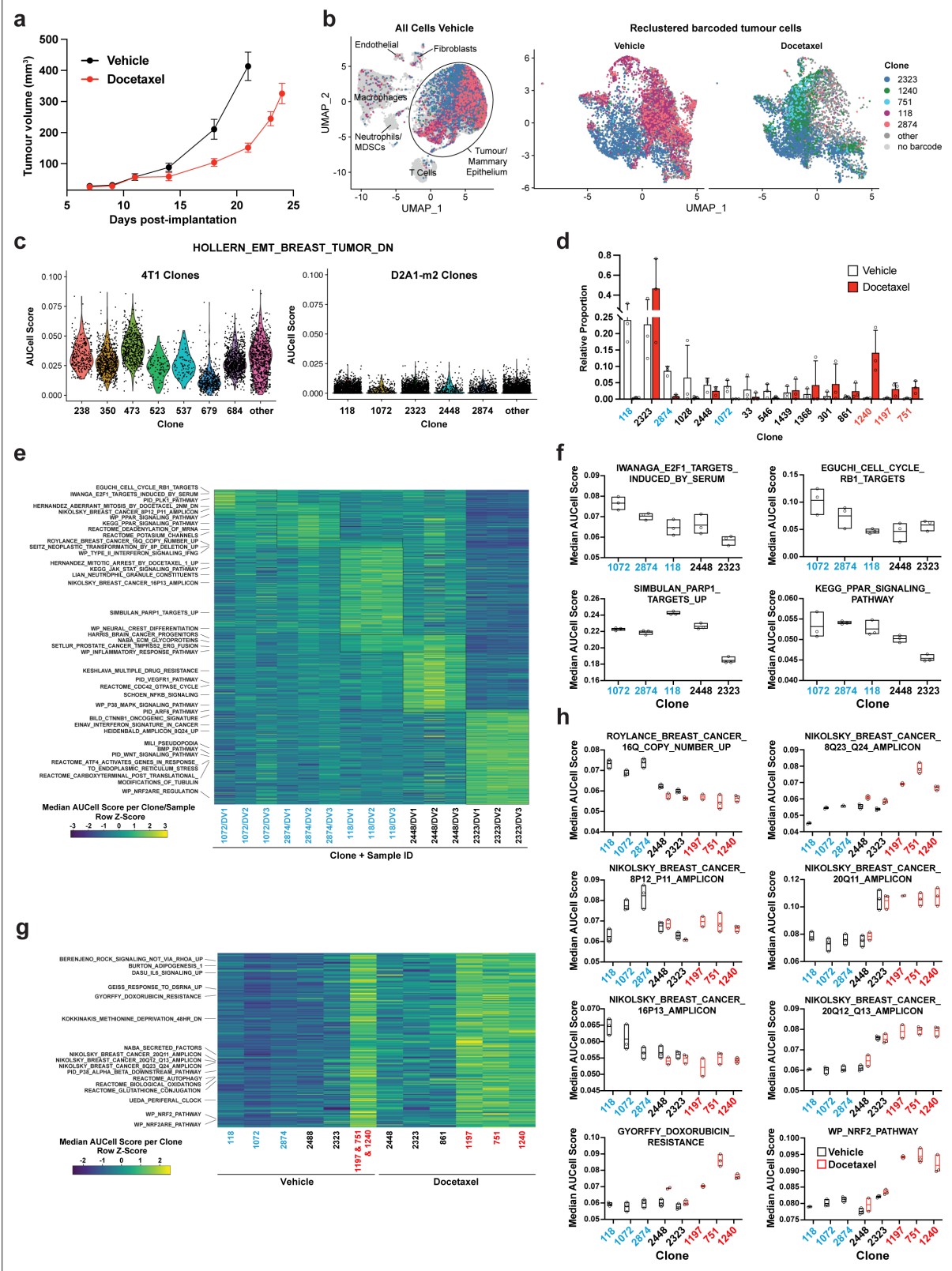

**Figure 4.** Clonal transcriptomic signatures of response and resistance to taxane chemotherapy in the D2A1-m2 mammary carcinoma model. (**a**) D2A1-m2 WILD-seq tumour growth curves with docetaxel treatment. D2A1-m2 WILD-seq tumours were treated with docetaxel or vehicle from 7 days post-implantation for 2 weeks (n=5 vehicle-treated mice, n=4 docetaxel-treated mice). Dosing was performed as in *Figure 3a* for 4T1 tumours. Data represents mean ± SEM. (**b**) scRNA-seq of docetaxel-treated D2A1-m2 WILD-seq tumours. UMAP plots of vehicle-treated D2A1-m2 WILD-seq tumours

*Figure 4 continued on next page*

*Figure 4 continued*

(left panel) and reclustered barcoded tumour cells (right panel) from vehicle- and docetaxel-treated tumours. Combined cells from 3 mice per condition are shown. Cells for which a WILD-seq clonal barcode is identified are shown as dark grey or coloured spots. Cells which belong to five selected clonal lineages are highlighted. (**c**) Comparison of EMT status of major 4T1 and D2A1-m2 WILD-seq clones. Violin plot of AUCell scores from vehicle-treated tumour cells generated using the HOLLERN_EMT_BREAST_TUMOR_DN (*Hollern et al., 2018*) gene set, a set of genes that have low expression in murine mammary tumours of mesenchymal histology. 4T1 WILD-seq clones exhibit varying levels of expression of this gene set, whereas D2A1-m2 WILD-seq clones have consistently low levels of expression of these genes. (**d**) Clonal representation. Proportion of tumour cells assigned to each clonal lineage based on the WILD-seq barcode (n=3 tumours per condition). Clones which make up at least 2% of the assigned tumour cells under at least one condition are plotted. The most sensitive clones to docetaxel treatment 118, 2874, and 1072 are highlighted in blue and the most resistant clones 1240, 1197, and 751 are highlighted in red. Data represents mean ± SD. (**e**) Clonal transcriptomic signatures from vehicle-treated tumours. Heatmap of median AUCell scores per sample for each of the five most abundant clones. All gene sets which showed consistent and statistically significant enrichment (combined fisher p-value <0.01 & mean log$_2$ enrichment >0.1) in at least one of these clones are illustrated. (**f**) Selected gene sets whose expression is associated with sensitivity to docetaxel. Median AUCell scores per sample for each of the five most abundant clones is plotted. (**g**) Transcriptomic signatures associated with resistance to docetaxel. For vehicle-treated tumours, resistant clonal lineages identified by barcodes 1197, 751, and 1240 were combined to have enough cells for analysis. Gene sets with significantly enriched expression in these resistant clones in vehicle-treated tumours were determined (adjusted p-value <0.01 and log$_2$ enrichment >0.1). A heatmap of median AUCell scores per clone, per condition of these resistance-associated gene sets is plotted. (**h**) Selected gene sets whose expression is enriched or depleted in resistant clones. Median AUCell scores per clone, per sample are plotted for samples with at least 20 cells per clone. Due to changes in clonal abundance with treatment, and our analysis cut-offs, some clones can only be assessed under vehicle- or docetaxel-treated conditions.

Three clones robustly increased their relative abundance within the tumour following docetaxel treatment, clones 1197, 751, and 1240, which despite making up less than 1% of the vehicle-treated tumours together constituted more than 20% of the docetaxel-treated tumours (*Figure 4d*). Due to the low abundance of cells in vehicle-treated samples, cells belonging to all three clones were pooled to analyse their baseline gene expression profiles (*Figure 4g*). Among the gene sets differentially expressed between resistant and sensitive clones, were a number of breast cancer amplicons indicating that there may be specific copy number variations associated with these clones (*Figure 4g and h*). However single-cell DNA sequencing data would be required to confirm the presence of specific genetic traits within our clones. Interestingly, gains in 8q24 (*Han et al., 2010*), 20q11 (*Voutsadakis, 2021*) and loss of 16q *Höglander et al., 2018* have previously been reported to be associated with resistance to taxane-based chemotherapy in agreement with our findings. Highly upregulated within all three of our resistant clones were genes related to the NRF2 pathway, even in the absence of docetaxel treatment (*Figure 4g and h*). NRF2 activation has been linked to cancer progression and metastasis and has been suggested to confer resistance to chemotherapy (*Homma et al., 2009*; *Jiang et al., 2010*; *Konstantinopoulos et al., 2011*; *Romero et al., 2017*; *Shibata et al., 2008*; *Singh et al., 2006*).

## Delineating the contribution of clonal abundance to gene expression changes upon drug treatment

Prior to the advent of single-cell sequencing, the majority of studies relied on bulk RNA-seq or microarray analysis of gene expression to interrogate the effect of chemotherapeutic interventions. While informative, these studies cannot differentiate between changes in bulk gene expression that arise due to clonal selection and changes that are induced within a clonal lineage as the result of drug exposure. Even with single-cell sequencing, definitive identification of the same clonal population across treatment conditions is impractical as it relies upon gene expression to group and cluster the cells, and if gene expression changes identifying the same subtypes of tumour cells with and without treatment is technically challenging. Our method alleviates these difficulties by enabling the direct comparison of clones of the same lineage under different conditions.

To examine the relative contribution of clonal selection and transcriptional reprogramming to changes in gene expression upon chemotherapy, we compared analysis of gene expression within each clone individually to a combined analysis of all pooled tumour cells (*Figure 5*, *Supplementary file 11*). Consistent with their mode of action, docetaxel had relatively little effect on the transcriptome of individual clones (*Figure 5a and b*) while JQ1 caused substantial changes to the transcriptome predominantly down-regulating gene expression (*Figure 5c*). Genes were identified under all treatments that were altered within the tumour as a whole but as a result of clonal selection rather than intra-clonal changes in gene expression, with the biggest effects being observed with docetaxel

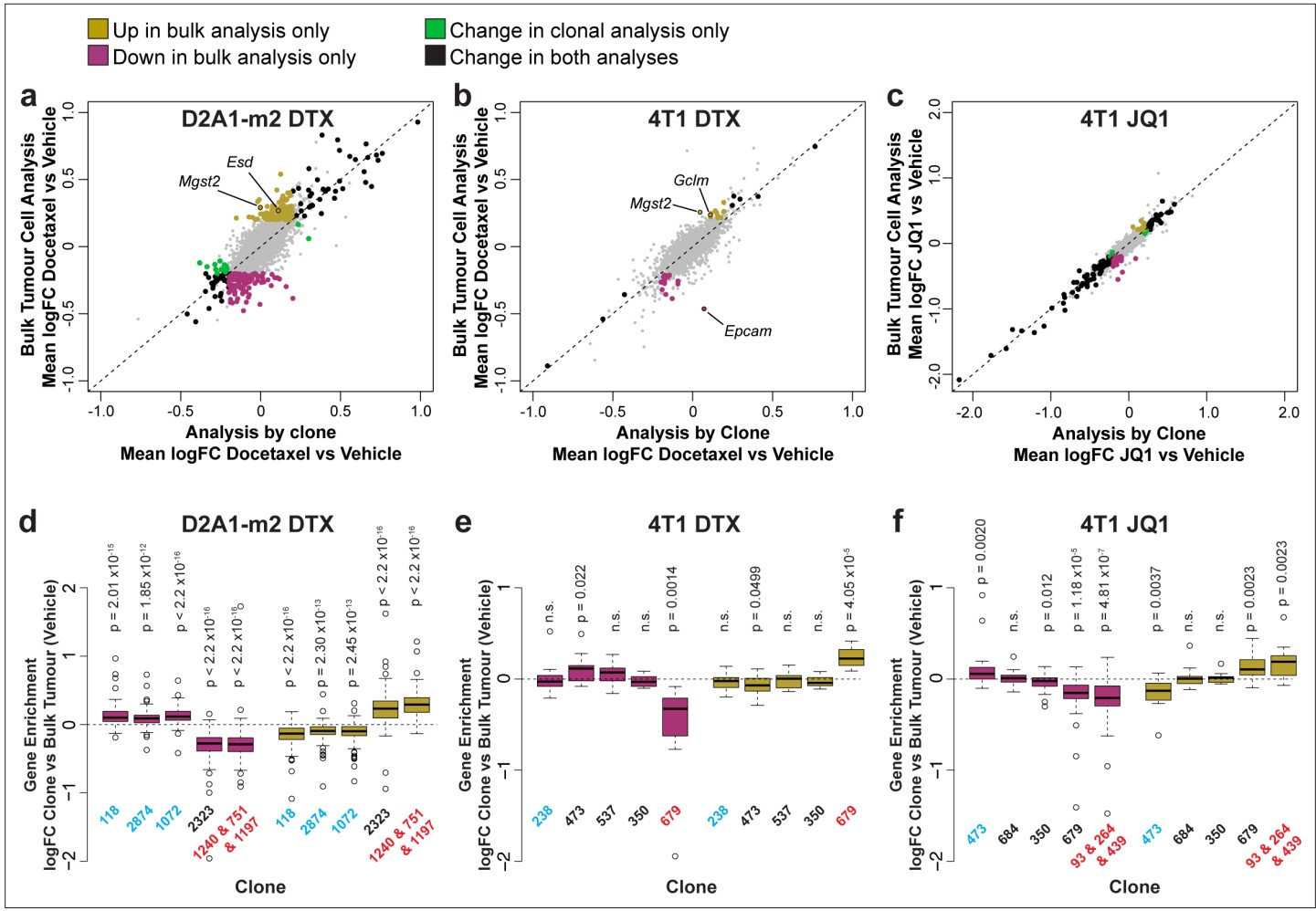

**Figure 5.** Delineating the contribution of clonal abundance to gene expression changes upon drug treatment. (**a and b and c**) Comparison of differential gene expression analysis in bulk tumour cells and intra-clonal changes in gene expression. Differential gene expression was performed for all barcoded tumour cells irrespective of clonal lineage comparing chemotherapy-treated and vehicle-treated cells (bulk tumour cell analysis). Alternatively differential gene expression was performed for each individual clone separately and the results combined to identify genes which robustly undergo intra-clonal changes in expression (analysis by clone). Whereas bulk tumour cell analysis will identify changes in overall gene expression due to both changes in clonal abundance and changes within the cells, analysis by clone enables us to delineate exclusively induced cellular changes in gene expression. Log$_2$ fold change in expression as determined by each of these analysis methods is plotted. Genes with significant changes in expression with chemotherapy (p-value <0.05, logFC <–0.2 or>0.2) are highlighted based on the method under which they were identified. Genes identified as significantly changing by one method only met neither logFC nor p-value cutoffs in the alternative method. (**d and e and f**) Changes in gene expression that are identified by bulk tumour cell analysis only can be attributed to changes in clonal abundance. The expression of genes which were identified as differential expressed after chemotherapy only in the bulk tumour cell analysis was assessed across clonal lineages at baseline. Baseline gene enrichment for each clone was determined as described previously by comparing cells of a specific clonal lineage to all barcoded tumour cells within the same vehicle-treated sample or experiment. Gene enrichment values for all genes with differential expression only in the bulk tumour cell analysis were plotted. As expected, genes down-regulated in bulk analysis have lower expression in resistant clones, whereas genes up-regulated in bulk analysis are enriched in resistant clones. p-values represent a one sample t-test vs a theoretical mean of 0.

treatment in D2A1-m2 tumours, in agreement with this condition inducing the largest changes in relative clonal abundance. To confirm that changes in gene expression detected in bulk tumour analysis but not the clonal analysis could be attributed to differences in clonal sensitivity to chemotherapy, we analysed baseline expression of these genes across the major clonal populations (***Figure 5d, e and f***). As expected, we found that genes upregulated only in bulk tumour analysis had significantly higher expression in clones resistant to chemotherapy (that increase in abundance with treatment) and genes only down-regulated in bulk tumour analysis had significantly lower expression in these resistant clonal lineages.

Among the genes that change in expression within the tumour as a whole as a result of clonal selection upon docetaxel treatment, we identified a number of genes related to glutathione synthesis and conjugation including *Mgst2*, *Esd*, and *Gclm* (*Figure 5a and b*), that may endow resistant clones with greater ability to resolve reactive oxygen species (ROS) induced by docetaxel (*Alexandre et al., 2007*). Of note, we also observed that in 4T1 tumours, *Epcam* was significantly reduced in expression in the bulk tumour but was not changed within the individual clonal populations (*Figure 5b*). This suggests that rather than inducing an EMT within the tumour cells, docetaxel is selecting clones of a pre-existing more mesenchymal phenotype.

## Convergent WILD-seq analysis across models identifies redox defense as a mediator of taxane resistance

To examine if there were any shared mechanisms of taxane resistance across our 4T1 and D2A1-m2 WILD-seq clones, we looked for genes that were enriched in resistant clonal lineages in both models. 4T1 resistance genes were defined as those that were significantly enriched in resistant clone 679 but not in sensitive clone 238 ($p<0.05$). D2A1-m2 resistance genes were defined as those that were significantly enriched in combined resistant clones 1240, 751, and 1197 but not in sensitive clones 118, 2874, or 1072 ($p<0.05$). In all cases, resistance genes were defined from vehicle treated tumours We identified 47 overlapping resistance genes (*Figure 6a*, *Supplementary file 12*). These genes were significantly enriched in pathways related to resolution of oxidative stress including the NRF2 pathway and glutathione-mediated detoxification (*Figure 6b*). Further analysis of D2A1-m2 and 4T1 clones confirmed that our 47 gene taxane resistance signature is enriched in the clones from which it was derived (D2A1-m2 clones 1197,1240, 751 and 4T1 clone 679) but strikingly also enriched in 4T1 clone 439 after docetaxel treatment which was not used to derive the signature (*Figure 6c*). 4T1 clone 439 exhibited resistance to docetaxel in our experiments (*Figure 3c and d*, *Figure 3—figure supplement 1*) but was not a clone we focused our analysis on as its low abundance in vehicle-treated tumours prevented us being able to define a basal transcriptomic signature. These data suggest that our 47 gene taxane resistance signature can de novo identify taxane resistant clones in our models.

Given the enrichment for genes related to NRF2 signaling in our resistance gene signature, we also assessed whether expression of NRF2-target genes was a predictor of docetaxel resistance across our two models (*Figure 6d*). Analysis using an NRF2-target gene set from the ChEA database (*Lachmann et al., 2010*) across clones from D2A1-m2 confirmed our prior observation (*Figure 4h*) that taxane-resistant clones 1197, 751 and 1240 are highly enriched for NRF2 signaling. Across 4T1 clones the only clone that showed enrichment for this gene set was clone 439. This 4T1 clone shows remarkable similarities to our D2A1-m2 resistant clones, in that it shows high expression of NRF2-targets, it has low abundance in vehicle-treated tumours and increases profoundly with docetaxel treatment. The most consistently resistant 4T1 clone 679 did not show overall enrichment for the complete NRF2-target gene set (*Figure 6d*) but was highly enriched for some key NRF2-targets involved in the oxidative stress response (*Figure 6—figure supplement 1*) which drive the enrichment in the overlapping list.

Importantly, the human orthologs of our 47 taxane resistance genes (*Figure 6e*, *Supplementary file 12*) as well as NRF2-target genes (*Figure 6f*) were strongly enriched in human patient tumours following combined anthracycline and taxane-based therapy, highlighting the potential clinical significance of our findings. Gene expression data from a previously published study with paired pre-neo adjuvant chemotherapy (NAC) core needle biopsies and post-chemotherapy surgical samples (*Vera-Ramirez et al., 2013*) were re-analysed using GSVA (*Hänzelmann et al., 2013*) to determine the effect of chemotherapy on a gene set composed of our 47 overlapping resistance genes (*Figure 6e*) as well as the ChEA NRF2-target gene set (*Lachmann et al., 2010*; *Figure 6f*). Expression of both these gene sets was significantly increased after chemotherapy, which our data would suggest is the result of outgrowth of resistant clonal lineages with increased propensity to withstand taxane-induced oxidative stress.

## Targeting taxane-resistant, NRF2-high clones through non-essential amino acid deprivation

Given our findings, we hypothesised that combining taxane-based chemotherapy with a drug specifically targeting resistant clones with high NRF2 signaling would provide a highly effective treatment regime. To test this hypothesis, we leveraged the finding that tumours with constitutively active NRF2,

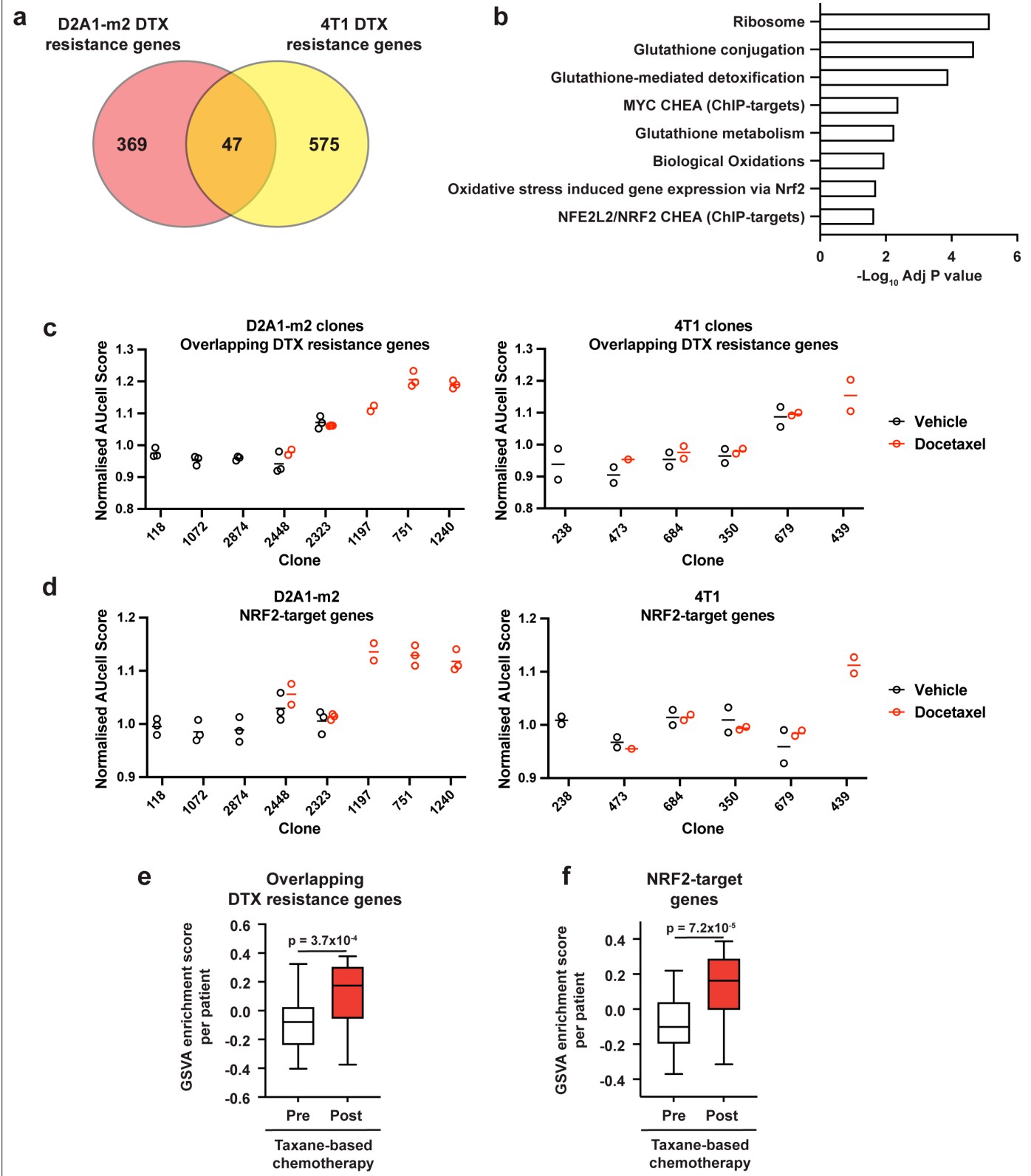

**Figure 6.** Taxane-resistant clones have elevated NRF2 signaling. (**a**) Overlap of genes associated with resistance between the D2A1-m2 and 4T1 WILD-seq models. 4T1 resistance genes were defined as those that were significantly enriched in resistant clone 679 but not in sensitive clone 238 (p<0.05). D2A1-m2 resistance genes were defined as those that were significantly enriched in combined resistant clones 1240, 751, and 1197 but not in sensitive clones 118, 2874, or 1072 (p<0.05). In all cases resistance genes were defined from vehicle treated tumours. (**b**) Gene set enrichment

*Figure 6 continued on next page*

*Figure 6 continued*

analysis of overlapping resistance genes. Gene set enrichment was performed using Enrichr for the human orthologs of the 47 overlapping resistance genes identified in a. Adjusted p-values for a subset of significant gene sets are plotted. (**c**) Our docetaxel resistance gene set identifies resistant clones. Our 47 overlapping docetaxel resistance genes were used as a gene set to calculate an AUCell expression score per clone, per sample (D2A1-m2) or per experiment (4T1). Normalised median AUCell scores are plotted for clones of interest. Data points were included if they represented at least 20 single cells. (**d**) NRF2-targets defined by ChIP have elevated expression in the most docetaxel resistant clones. NRF2 targets were taken from the ChEA (ChIP Enrichment Analysis) database and their expression measured across clones of interest. Normalised median AUCell scores are plotted for clones of interest. (**e**) Expression of our identified resistance genes is increased in human breast tumours following docetaxel treatment. Expression of our 47 overlapping resistance genes was assessed in human breast cancer samples taken before and after taxane-based neoadjuvant chemotherapy (GSE28844). GSVA enrichment scores for our gene set was calculated for samples from 28 patients for which matched pre- and post-treatment gene expression data were available. Patients received one of three taxane-containing treatment regimens; Regimen A: Epirubicin 90 mg/m²-Cyclophosphamide 600 mg/m², 3 cycles bi-weekly and *Paclitaxel* 150 mg/m²-Gemcitabine 2500 mg/m², 6 cycles bi-weekly ±weekly Herceptin 4 mg/Kg during the first week, 2 mg/Kg for the remaining 11 cycles. Regimen B: Doxorubicin 60 mg/m²-Pemetrexed 500 mg/m², 4 cycles tri-weekly and *Docetaxel* 100 mg/m², 4 cycles tri-weekly. Regimen C: Doxorubicin 60 mg/m²-Cyclophosphamide 600 mg/m², 4 cycles tri-weekly and *Docetaxel* 100 mg/m², 4 cycles tri-weekly. Expression of our overlapping resistance gene set was significantly increased after chemotherapy in human samples. p-value calculated by paired t-test. (**f**) NRF2-target genes are upregulated in human patients following neoadjuvant chemotherapy. GSVA enrichment scores for NRF2-target genes (NFE2L2 CHEA consensus CHIP-targets) were calculated for samples from 28 patients in the GSE28844 dataset for which pre- and post-treatment gene expression data were available. p-values calculated by paired t-test.

The online version of this article includes the following figure supplement(s) for figure 6:

**Figure supplement 1.** Expression of bona fide transcriptional targets of NRF2 involved in ROS detoxification (*Gstm2*, *Mgst2*, *Mgst1*) and glutathione production (*Gclc*, *Gclm*) (related to **Figure 6d**).

due to mutation in the negative regulator *Keap1*, have metabolic vulnerabilities that arise from their high antioxidant production (***Romero et al., 2017***), including dependency of glutamine (***Romero et al., 2017***) and a general dependency on exogenous non-essential amino acids (NEAA) including asparagine (***LeBoeuf et al., 2020***). This metabolic dependency can be targeted therapeutically using L-asparaginase (ASNase from *E. coli*), which is deployed in the clinical management of acute lympho-blastic leukemia (*ALL*) (***Batool et al., 2016***), and catalyzes the conversion of asparagine to aspartic acid and ammonia (***Chan et al., 2019***).

To ascertain whether docetaxel-resistant clones were collaterally sensitive to ASNase, we treated D2A1-m2 WILD-seq tumours initially with docetaxel to select for resistant clones and then began daily treatment with L-asparaginase one week later. This dosing regime was chosen as we found that with the dose of docetaxel used in this study, co-administration of the 2 drugs or treatment with ASNase immediately following docetaxel was poorly tolerated. As shown in ***Figure 7a***, treatment with ASNase arrested tumour growth and led to a ~40% increase in time to endpoint (relative to vehicle) in this highly aggressive model, although the tumours did acquire resistance and regrew after approximately one week of treatment. Importantly, ASNase alone had no significant effect on tumour growth (***Figure 7b***), suggesting that docetaxel-dependent expansion of NRF2-high clones is required for ASNase treatment to induce a change in bulk tumour growth. To determine the response of individual clonal lineages within the bulk tumour to ASNase treatment, we performed single cell RNA sequencing on vehicle treated tumours (day 21), as well as docetaxel treated tumours before the start of ASNase treatment (day 21) and after 4 doses of ASNase (day 25). As before, our docetaxel-resistant clones, 751, 1197 and 1240, which have high levels of NRF2 signaling all exhibited a dramatic increase in their abundance with docetaxel treatment (***Figure 7c***). Excitingly, clones 751 and 1197 were sensitive to ASNase returning to baseline levels. Clone 1240 decreased in abundance in 2 of the 3 mice analysed so is likely to also be sensitive to ASNase and further experiments confirm this (see below). As predicted, our NRF2-high resistant clones were selectively targeted by amino acid deprivation as other clones such as 2323 were unchanged in their relative abundance (***Figure 7c*** and ***Figure 7—figure supplement 1a***).

To confirm the mechanism of action of L-asparaginase and identify potential mechanisms of resistance to this drug that might cause the relapse observed, we analysed the transcriptomic effects of asparagine deprivation on docetaxel pre-treated tumours before and after ASNase administration. Genes which consistently changed in expression after ASNase treatment across clonal lineages are shown in ***Figure 7d***. Many of the genes found to be differentially expressed in our tumour cells following L-asparaginase treatment are either directly related to protein synthesis (*Eif3c*, *Gars*, *Eif3g*, *Eif5a*) or are consistent with changes in gene expression reported in cell lines following amino acid

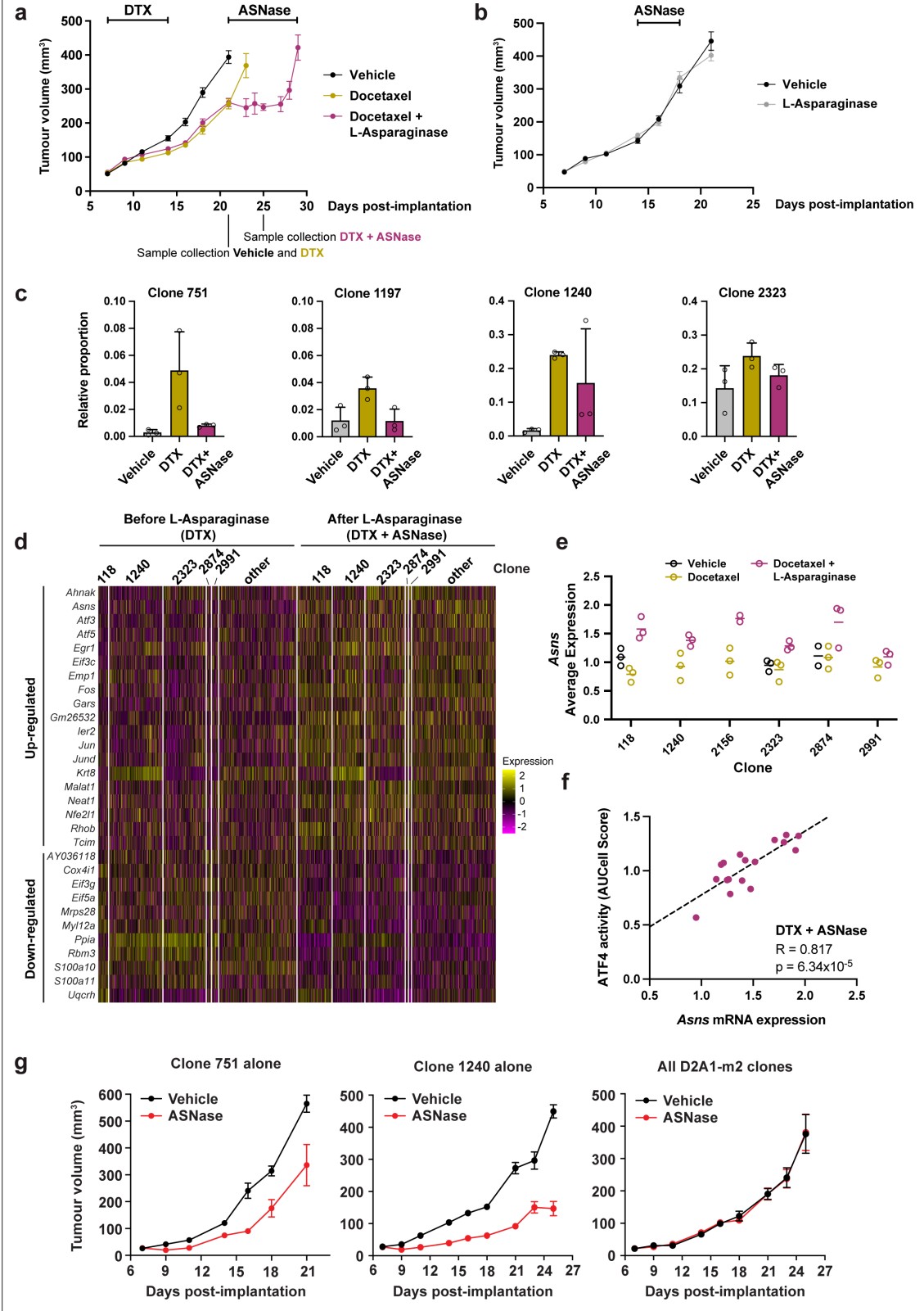

**Figure 7.** Taxane-resistant, NRF2-high clones are inherently sensitive to amino acid deprivation. (**a**) Docetaxel-resistant tumours are collaterally sensitive to L-asparaginase. D2A1-m2 WILD-seq tumours were treated with 3 doses of 12.5 mg/kg docetaxel (days 7,9,11 post-implantation) and 1 dose of 10 mg/kg docetaxel (day 14 post-implantation). From day 21 mice were treated daily with L-asparaginase. Indicated below the X axis are the timepoints of tumour collection for single-cell sequencing. Measurements are combined from two independent experiments, error bars represent SEM. Due to

*Figure 7 continued on next page*

*Figure 7 continued*

sample collection at timepoints indicated the number of animals is reduced beyond this. Vehicle n=15 mice, docetaxel n=14 mice (reduced to 5 mice after day 21), docetaxel + L-asparaginase n=13 mice (reduced to 4 mice from day 25). In addition, 2 mice reached humane endpoint (due to weight loss following docetaxel treatment but prior to administration of L-asparaginase) one in the DTX only arm at day 18 and one in the DTX +L Asp arm at day 21. (**b**) L-asparaginase alone does not affect tumour growth. D2A1-m2 WILD-seq tumours were treated with L-asparaginase or vehicle for 5 consecutive days from day 14 post-implantation. n=10 mice per condition, error bars represent SEM. (**c**) Taxane-resistant clones are sensitive to L-asparaginase. Relative clonal abundance in vehicle-treated (day 21), docetaxel-treated (day 21) and docetaxel and L-asparaginase-treated (day 25) D2A1-m2 WILD-seq tumours is shown for 3 taxane-resistant clones (751, 1197, 1240) and 1 neutral clone (2323). Clonal proportions were calculated from single cell sequencing data of 3 tumours per condition. Error bars represent SD. (**d**) Gene expression changes in tumour cells after L-asparaginase treatment. Heatmap for genes which are most significantly and consistently differentially expressed across clonal lineages after treatment with L-asparaginase. 2400 cells are represented (400 per sample), grouped according to clonal lineage. (**e**) *Asns* expression increases after L-asparaginase treatment. *Asns* expression was calculated per clone, per sample using the AverageExpression function from the Seurat package. Data points were included if they represented at least 30 single cells and there were at least 2 data points per condition. (**f**) *Asns* expression correlates with ATF4 activity in tumours treated with the docetaxel -L-asparaginase combination. ATF4 transcriptional activity was calculated using a gene set defined by Tameire et al. but from which *Asns* was removed. Each data point represents the normalised, median AUCell score for a specific clone and animal. Only animals treated with docetaxel and then L-asparaginase were included. Correlation was determined using the Pearson correlation test. (**g**) Clone 751 and 1240 are inherently sensitive to L-asparaginase. The effect of L-asparaginase treatment on tumour growth of monoclonal tumours composed only of clone 751 (250,000 cells implanted) or clone 1240 (250,000 cells implanted) was compared to that of tumours established from the heterogeneous WILD-seq D2A1-m2 pool (60,000 cells implanted). Mice were treated with L-asparginase (60 U/day, IP, 5 days a week) from 7 days post-implantation. Significance was determined by two-way anova (clone 751: $F_{(1,8)}$ = 18.24, p=0.0027; clone 1240: $F_{(1,8)}$ = 108.8, p=$6.193 \times 10^{-7}$, all clones: $F_{(1,7)}$ = 0.00014, p=0.991). n=5 animals, except for L-asparaginase-treated D2A1-m2 pool where n=4.

The online version of this article includes the following figure supplement(s) for figure 7:

**Figure supplement 1.** Clonal representation (related to *Figure 7c*) and lack of significant correlation between Asns expression and ATF4 activity in the absence of L-asparaginase treatment.

**Figure supplement 2.** Validation of specific WILD-seq barcode detection by qPCR.

deprivation including *Atf5*, *Atf3*, *Jun*, *Fos*, *Egr1*, and *Asns* (***Fu et al., 2011***; ***Pan et al., 2003***; ***Pohjan-pelto and Hölttä, 1990***; ***Shan et al., 2010***). Of specific interest is the up-regulation of asparagine synthetase (ASNS) (***Figure 7e***) which catalyses the de novo biosynthesis of L-aspartate and is therefore a potential mediator of ASNase resistance. In acute lymphoblastic leukemia (*ALL*), low levels of ASNS resulting in a dependence on extracellular asparagine are considered an important biomarker for L-asparaginase treatment. Moreover, the importance of ASNS overexpression in conferring asparaginase resistance has been well documented and is frequently seen in *ALL* patients that develop drug-resistant forms of the disease following treatment with ASNase (reviewed in ***Richards and Kilberg, 2006***). The stress responsive transcription factor ATF4 is a well-known regulator of ASNS expression under conditions of amino acid deprivation. To determine whether ATF4 activity was associated with *Asns* induction in our model, we utilised a gene set indicative of ATF4 transcriptional activity, from which we removed *Asns* itself. AUCell analysis revealed a strong correlation between *Asns* and ATF4 activity across clones in ASNase treated tumours (***Figure 7f***. *R*=0.817, p=$6.3 \times 10^{-5}$) but not in vehicle (*R*=0.131, p=0.758) or DTX (*R*=0.397, p=0.158) treated tumours (***Figure 7—figure supplement 1b***), suggesting that ATF4 specifically drives asparaginase-induced *Asns* expression. In our experiments, baseline *Asns* expression was consistent across clones and induction in response to asparaginase was observed across all clones analysed, albeit to slightly different magnitudes, (***Figure 7e***) suggesting a general resistance mechanism and supporting the clinical utility of an ASNS inhibitor, if one were to be developed, as third line treatment in this context.

We hypothesised that the observed effects of ASNase on our taxane-resistant clones was an inherent property of these clones rather than a docetaxel-induced phenomenon, since they exhibit basally high levels of NRF2 signalling. However, due to the low abundance of these clones prior to docetaxel treatment (<1%) accurate estimation of the effect of ASNase alone on these clones in the absence of docetaxel-induced selection is challenging. As an alternative means to test this hypothesis, we attempted to establish monoclonal cell lines of these clones by isolating them from their parental pool by single-cell cloning. Single-cell clones were screened by qPCR of the genomic DNA with a barcode-specific reverse primer and a forward primer that binds with the zsGreen transgene (for primer validation see ***Figure 7—figure supplement 2***). Using this strategy, we successfully isolated the NRF2-high clones 751 and 1240. We then implanted these clones individually into BALB/c mice and after one week we initiated treatment with ASNase monotherapy. As shown in ***Figure 7g***, monoclonal

tumours derived from clone 751 or 1240 both exhibited significant suppression of tumour growth in response to ASNase (*Figure 7g*), while tumours derived from our standard 3 pools of D2A1-m2 WILD-seq cells containing all clones (*Figure 7g*) did not. These data strongly indicate that NRF2-high clones have a pre-existing sensitivity to ASNase and that docetaxel treatment merely increases their abundance rather than shaping their phenotype.

In summary, our data support the notion that WILD-seq can identify causal mechanisms of drug resistance in vivo, that can be leveraged to inform new combination therapies. Since the redox defense signatures we identified are detectable in patients after neo-adjuvant chemotherapy (NAC), one can envisage an approach whereby patients receiving NAC have the surgical tumour specimen profiled for NRF2 gene signatures and those with high levels receive a post-operative course of L-asparaginase.

## Discussion

Tumour heterogeneity is thought to underlie drug resistance through the selection of clonal lineages that can preferentially survive therapy. However, identifying the features of such lineages, so that they can be targeted therapeutically, has been challenging due the lack of understanding of their molecular characteristics and the lack of animal models to prospectively test therapeutic interventions and combinations thereof. To overcome these challenges, we developed WILD-seq, a system that leverages expressed barcodes, population bottle necking, syngeneic mouse models and single cell RNA-seq to link clonal lineage to the transcriptome. Among the existing methods for coupling lineage tracing with single cell transcriptomic profiling, the majority use either lentiviral delivery of a genetic barcode similar to that used here or CRISPR/Cas9-mediated mutations for clonal lineage identification (*Biddy et al., 2018*; *Gutierrez et al., 2021*; *Quinn et al., 2021*; *Simeonov et al., 2021*; *Weinreb et al., 2020*). We chose to avoid CRISPR/Cas9-based lineage labelling as induction of DNA damage could have an impact on the transcriptome and the sensitivity of the cells to therapeutic agents (*Haapaniemi et al., 2018*; *Jiang et al., 2022*) many of which cause DNA damage as part of their mechanism of action. Our approach is unique in that we purposefully bottleneck our clonal population to achieve a balance between maximising clonal diversity and minimising variation in clonal representation across replicate animals and experiments. It is this feature that allows us to robustly call clonal gene expression signatures and differential clonal abundance before and after therapeutic intervention and it is this in turn that allows us to identify relevant drug resistance mechanisms in vivo.

Single cell RNA-seq is a powerful tool to study complex biological systems and recent analytical advances have facilitated the integration of diverse datasets to infer effects of perturbations. While these approaches are well suited to integrating data across samples and conditions at the level of cell types which are discriminated by large differences in gene expression, they are currently unable to reliably identify the same subtypes of tumour cells, such as different clonal lineages, across samples as these are discriminated by relatively small differences in gene expression and this data is confounded by gene expression changes induced by the local microenvironment. These issues are further exacerbated when the perturbation under investigation induces further and potentially clonal lineage-specific transcriptomic changes. Barcoding approaches, such as WILD-seq, circumvent these problems by using the barcode to match sub-populations of cells across conditions and treatments without reliance on their transcriptomic profile and thereby enables the deconvolution of changes in abundance of subtypes of tumour cells and changes in the cell state. As such, barcoding with WILD-seq facilitates interpretation of perturbations in single-cell RNA-seq regardless of whether changes in clonal abundance are an expected outcome.

We find that the abundance of clones in cell culture and in vivo differ greatly, with the most abundant clones in vitro being lowly represented in vivo and vice versa thus providing a cautionary note when analyzing drug response in vitro. Moreover, WILD-seq of 4T1 tumours revealed that the relative immune competence of the host profoundly sculpts the transcriptome of clonal lineages and, as exemplified by JQ1, therapeutic interventions can impact the tumour microenvironment and its interaction with tumour cells, effects that would be missed in vitro and in immunocompromised hosts. We utilised WILD-seq to analyze sensitivity and resistance to taxane chemotherapy in two syngeneic, triple negative, mammary carcinoma models highlighting both known and new pathways of resistance (*Marine et al., 2020*). Resistance to cancer therapies can arise due to clonal selection or through adaptive reprograming of the epigenome and transcriptome of individual clones. Our data with docetaxel treatment in the 4T1 and D2A1-m2 models indicate that, over the time frames we

have examined, clonal selection is the dominant force driving resistance to taxane chemotherapy, with gene expression signatures, such as EMT and NRF2 signaling, being present in clones at baseline that are then selected for during therapy. Critical support for a clonal selection mechanism comes from the observation that monoclonal tumours derived from NRF2-high D2A1-m2 clones are inherently sensitive to L-asparaginase without ever having been exposed to taxanes, indicating that docetaxel merely enriches for these clones that have a pre-existing vulnerability to asparagine deprivation. Conversely, up-regulation of *Asns*, which was detected across clonal lineages after L-asparaginase treatment, is suggestive of an adaptive transcriptional resistance phenotype. Therefore, depending on the mode of action of specific drugs, transcriptional reprogramming may also induce therapeutic resistance and such mechanisms can also be effectively identified with the WILD-seq platform where clonal resolution can confirm a lack of clonal selection. For example, it has been proposed that tumour cells can dynamically switch between epithelial and mesenchymal states (*Shibue and Weinberg, 2017*) and that such cells may transition to a more mesenchymal state to resist the initial therapy and then transition back towards a more epithelial state to proliferate and repopulate the tumour. While the data presented here suggest that the shift towards a more mesenchymal tuour phenotype upon taxane treatment is driven by clonal selection rather than a change in the state of cells, it is entirely plausible that transient changes in EMT status occur within our clonal population that are missed by sampling at endpoint. Future experiments with sampling of multiple time points and/or re-transplantation experiments present an interesting future application of the WILD-seq system that could directly address these questions where barcoding would be critical to differentiate between changes in cell state and changes in abundance.

Applying WILD-seq to examine docetaxel response across two TNBC models afforded the opportunity to overlap resistance genes for the same drug across models and remove model-specific effects. These analyses uncovered a critical role for redox defense in docetaxel resistance that also appears to be operative in human breast cancer patients after chemotherapy. Having identified a primary cause of resistance, we next sought to explore the possibility of collateral sensitivity. Collateral sensitivity, first described for antibiotics (*Imamovic and Sommer, 2013*; *Pluchino et al., 2012*; *Roemhild and Andersson, 2021*) is the phenomenon by which resistance to one drug comes at the cost of sensitivity to a second drug. In the context of cancer and taxanes, collateral sensitivity has the distinct advantage over other therapeutic strategies of maintaining the initial first line therapy and only modifying subsequent therapies. We took advantage of previous findings linking constitutive NRF2 signaling, via *Keap1* loss, to a dependency on exogenous non-essential amino acids (*LeBoeuf et al., 2020*) and thereby sensitivity to L-asparaginase. Application of L-asparaginase after docetaxel treatment led to an initial cessation of tumour growth followed by regrowth 6 days later. WILD-seq of docetaxel-treated tumours before and after L-asparaginase treatment confirmed the specific suppression of NRF2-high clones and also revealed a compensatory, largely clone agnostic, up-regulation of asparagine synthetase (*Asns*), which likely drives relapse in these tumours given the importance of ASNS to L-asparaginase resistance in *ALL* (*Richards and Kilberg, 2006*). Interestingly, we have previously shown that asparagine bioavailability regulates EMT and metastatic progression in breast cancer models (*Knott et al., 2018*). Thus, asparagine deprivation, which has not been extensively explored in breast cancer, may present multiple benefits to patients and the utility of L-asparaginase, a clinical stage drug, in this setting warrants further investigation.

This study highlights the challenges of tackling tumour heterogeneity therapeutically. Even though we can effectively suppress the induction of docetaxel resistant clones by administration of L-asparaginase the tumours still adapt to this intervention and regrow, most likely due to transcriptionally shifting their metabolism towards de novo asparagine synthesis. Nevertheless, hope still remains since there are only three avenues by which cells can supply themselves with asparagine (1) uptake of extracellular asparagine which is effectively shut-off by ASNase (2) de novo synthesis through Asns or (3) catabolism of existing proteins. If we could effectively force tumours to depend on synthesis through ASNS, we could then deprive them of that additional dependency if ASNS-directed therapeutics were to be developed. This concept of steering clonal evolution with drugs towards a predictable and irreconcilable, therapeutically targetable, dependency may provide a general approach to achieving durable therapeutic responses for which tractable models of tumour evolution, such as those described here, are essential predictive components.

# Materials and methods

**Key resources table**

| Reagent type (species) or resource | Designation | Source or reference | Identifiers | Additional information |
|---|---|---|---|---|
| Strain, 6–8 week old, female, Balb/C (*Mus musculus*) | Balb/C | Charles River | Strain code 028 | |
| Strain, 6–8 week old, female, NOD.Cg-Prkdcscidll2rgtm1Wjl/SzJ (*Mus musculus*) | NOD scid gamma (NSG) | Charles River | Strain code 614 | |
| Cell line (*Mus musculus*) | 4T1 | ATCC | Cat # CRL-2539. RRID:CVCL_0125 | |
| Cell line (*Mus musculus*) | D2A1-m2 | Prof Clare Isacke's laboratory. *Jungwirth et al., 2018* | | Derived from in vivo selection of D2A1 parental line from the metastatic site (lung) |
| Cell line (*Homo sapiens*) | 293 FT | Thermo Fisher Scientific | RRI:CCVCL_6911 | Used to make lenti-virus |
| Recombinant DNA reagent | pHSW8 | This paper | | Empty WILD-seq vector with zsGreen flourophor. See materials and methods. |
| Recombinant DNA reagent | WILD-seq vector library | This paper | | WILD-seq vector barcode library in the 3' UTR of szGreen. See materials and methods. |
| Commercial assay or kit | SYBR green PCR master mix | Applied Biosystems/Thermo Fisher Scientific | Cat # 4309155 | |
| Commercial assay or kit | Gibson Assembly master mix | New England Biolabs | Cat # E2611S | |
| Commercial assay or kit | High pure RNA isolation kit | Roche | Cat # 11828665001 | |
| Commercial assay or kit | Tumor dissociation kit, mouse | Miltenyi Biotec | Cat # 130096730 | |
| Commercial assay or kit | Chromium Single Cell 3' Reagent Kits User Guide (v3.1 Chemistry Dual Index) | 10x Genomics | User guide reference: CG000315 | |
| Chemical compound, drug | (+)-JQ1 | Selleck Chemicals | S7110 | |
| Chemical compound, drug | docetaxel | Selleck Chemicals | S1148 | |
| Chemical compound, drug | Native *E coli* L-asparaginase protein | Abcam | ab277068 | |
| Software, algorithm | Bartender | *Zhao et al., 2018* | | |
| Software, algorithm | DNA barcodes | *Buschmann, 2017* | | |
| Software, algorithm | Cell Ranger | 10x Genomics | RRID:SCR_017344 | |
| Software, algorithm | Seurat | *Stuart et al., 2019* | RRID:SCR_016341 | |

## Cell lines and culture

The mouse mammary tumor cell lines 4T1 (ATCC Cat# CRL-2539, RRID:CVCL_0125) and D2A1-m2 (a kind gift from Clare Isacke's lab *Jungwirth et al., 2018*) and the 293 FT (Thermo Fisher Scientific, RRID:CVCL_6911) packaging cell line for virus production were cultivated in DMEM high glucose (Gibco), supplemented with 10% heat-inactivated fetal bovine serum (Gibco) and 50 U/mL penicillin-streptomycin (Gibco). All cell lines were STR profiled to confirm their identity and tested negative for mycoplasma.

## Single-cell clone isolation

D2A1-m2 cells were thawed and cultured from the original pool (pool 3) of cells containing the docetaxel resistant clones (clone 751 and clone 1240). Single zsGreen-positive cells were then FACS

sorted into three 96-well plates. When passaged, duplicate 96-well plates were cultured. Cells from each row of the duplicate 96-well plates were pooled and gDNA extraction of each pooled row of cells was carried out using the Maxwell RSC instrument (Promega) and the 'Cultured cells DNA extraction kit' (Promega). These samples were then subjected to qPCR (Sybr green master mix, applied biosystems) using a barcode-specific reverse primer and a forward primer within zsGreen (see table below for qPCR primer sequences). These primer sequences were first validated by performing qPCR of gDNA derived from the three in vitro pools where we know which original 250 cell pool they should be present in (*Supplementary file 7c*) normalised to total zsGreen qPCR signal using the delta Ct method. Following identification of rows with a positive qPCR signal, when passaged, the corresponding cultured wells of the 96-well plate were individually seeded into 12-well plates. During the passage of these individual clonal pools, 250,000 cells were taken for gDNA extraction and qPCR analysis to confirm which wells contained a pure population of D2A1-m2 docetaxel resistant clone 751 or 1240.

| Name | Sequence |
| --- | --- |
| zsGreen_qPCR_fwd | CGTGTTCACCGAGTACCCC |
| zsGreen_qPCR_rev | ACGCCGTAGAACTTGGACTC |
| common_BC_qPCR_fwd | GAACCAGAAGTGGCACCTGAC |
| 2323_BC_qPCR_rev | CAAAGTTCTATCCGCTTCATAATGGC |
| 118_BC_qPCR_rev | CAAAGTTCTATCCGAGGCATACAGTA |
| 2874_BC_qPCR_rev | CAAAGTTCTATCCGAGTTACGATAGG |
| 1240_BC_qPCR_rev | AAAGTTCTATCCGTTAGAGTTGCGC |
| 1197_BC_qPCR_rev | GTTCTATCCGCAGGCTATTCGG |
| 751_BC_qPCR_rev | TTCTATCCGTGCCGAGCATTG |

Q-PCR reactions consisted of 12.5 µl Sybr Green Master Mix, 0.5 µl Fwd primer (10 µM), 0.5 µl Rev primer (10 µM), 2.5 µl gDNA (50 ng/µl) and PCR was performed using the following conditions: 95 °C 15 s, 60 °C 1 min, for 40 cycles. On a Bio-Rad C1000 Thermal Cycler.

## Virus production

The WILD-seq library was packaged using 293 FT lentivirus packaging cells. Cells were plated on 15 cm adherent tissue culture plates (Corning) one day before transfection at a confluency of ~70%. Lentiviral particles were produced by co-transfecting 293 FT cells with the transfer plasmid (32 µg) and standard third-generation packaging vectors pMDL (12.5 µg), CMV-Rev (6.25 µg) and VSV-G (9 µg) using the calcium-phosphate transfection method (Invitrogen). The transfection mixture was added to the packaging cells along with 100 mM chloroquine (Sigma-Aldrich). After 16–18 hr, media was replaced for fresh growth media. Viral supernatant was collected 48 hr after transfection and filtered through a 45 µm filter. The viral supernatant was applied directly to cells or stored at 4 °C for short-term storage or –80 °C for long-term storage. When necessary, virus was concentrated using ultra-centrifugation. Lentiviral titre was determined by serial dilutions and measurements of fluorescence via flow cytometry.

## WILD-seq library design and cloning

The pHSW8 lentiviral backbone was constructed using a four-way Gibson Assembly (NEB) by inserting a reverse expression cassette, consisting of an attenuated PGK promoter lacking 47 bp (see *Supplementary file 15*, pHSW8 vector sequence), the zsGreen ORF, a cloning site for high-diversity barcode libraries and a synthetic polyA signal, into an empty pCCL-c-MNDU3-X backbone (#81071 Addgene). To generate the WILD-seq library, a barcode cassette was introduced at the cloning site within the pHSW8 lentiviral backbone, using PCR (Q5 High-Fidelity DNA Polymerase, NEB) and Gibson Assembly (NEB), such that it is expressed within the 3'UTR of the zsGreen transcript.

| Name | Sequence |
|---|---|
| Assembly_Fwd | 5'-AAACTCTTGAGTGAACTCCAGTGATTTTGAACCAAGCGATTCAAAGTTCT-3' |
| Assembly_Rev | 5'-ccttgccctgaTAACTGGAGGCAGTAATTTACAGCCATGCGCTCGTTTAC-3' |
| BarcodeOligo_Fwd | 5'-TGAACCAAGCGATTCAAAGTTCTATCCGNNNNNNNNNNNNNtgcatcggttaaccgatgca-3' |
| BarcodeOligo_Rev | 5'-ATGCGCTCGTTTACTATACGATNNNNNNNNNNNNNtgcatcggttaaccgatgca-3' |

The barcode library was designed by generating 12 nt variable sequences using the R package DNABarcodes (*Buschmann, 2017*) and a set Hamming distance of 5. The resulting pool of sequences was then purchased as a custom oligo pool (Twist Bioscience). Reverse complement oligos (BarcodeOligo_Fwd/Rev) each containing a specific PCR handle, a 12 bp variable region and 20 bp constant linker were annealed and amplified by PCR for 20 cycles (using Assembly_Fwd/Rev primers). The amplified barcode library was column purified (Gel extraction kit, Qiagen) and the vector backbone was prepared by digestion with SwaI (NEB). WILD-seq barcodes were inserted into the lentiviral vector backbone through Gibson Assembly (NEB), concentrated and transformed into 10b electrocompetent *E. coli* cells (NEB).

## Bottlenecking strategy and characterisation of WILD-seq pools

4T1 or D2A1-m2 cells were infected with WILD-seq library at low MOI (~0.2–0.3). Two days after infection, the desired number of zsGreen positive cells, ranging from 10 to 1250 cells, were collected and cultured for 2 weeks to allow for the pool of clones to stabilize. Different pooling strategies were tested, the ultimate WILD-seq pool was generated from three independent pools each established from 250 sorted cells, maintained separately and mixed in equal proportions immediately prior to implantation.

## Library complexity analysis

WILD-seq barcodes of the lentiviral library were amplified using a one-step PCR protocol. 1 ng plasmid was used as template in three separate PCR reactions to account for PCR biases and errors. All reactions were pooled, concentrated and purified on a column and then sequenced on one lane of HiSeq4000. Reads that contained the WILD-seq barcode motif were identified and extracted from the FASTQ files. Detected WILD-seq barcodes were clustered by hamming distance using the Bartender algorithm (*Zhao et al., 2018*) and the most highly represented barcode sequence for each cluster selected.

## Whitelist generation of WILD-seq barcodes

To generate a comprehensive whitelist of expressed barcodes in each pool, RNA was extracted from WILD-seq transduced cells (High Pure RNA isolation kit, Roche) and reverse transcribed using the Superscript IV reverse transcription kit (Invitrogen) and a target site-specific primer with a unique molecular identifier (UMI) and an Illumina sample index. cDNA was amplified by PCR (Q5 High-Fidelity DNA Polymerase, NEB) using primers (RTWhitelist_Fwd/Rev) containing Illumina-compatible adapters. Alternatively, 1 µg of gDNA was extracted from WILD-seq transduced cells (Blood&Cell Culture DNA Kit, Qiagen) and the barcode amplified by PCR using primers containing Illumina-compatible adapters (gDNAWhitelist_Fwd/Rev). PCR products were purified via gel extraction (Qiagen) and quantified by Qubit. The library was sequenced on an Illumina MiSeq with a custom sequencing primer for Read1 (CustomRead1).

| Name | Sequence |
|---|---|
| RT Primer | 5'-CAAGCAGAAGACGGCATACGAGAT**NNNNNNN**GTGACTGGAGTTCAGACGTGTGCTCTTCCGATCTNNNNNNNNNCAAGCGATTCAAAGTTCTATCCG-3' |
| RTWhitelist_Rev | 5'-CAAGCAGAAGACGGCATACGA-3' |
| RTWhitelist_Fwd | 5'-AATGATACGGCGACCACCGAGATCTACACCAGCAGTATGCATGCGCTCGTTTACTATACGAT-3' |

*Continued on next page*

*Continued*

| Name | Sequence |
| --- | --- |
| gDNAWhitelist_Fwd | 5'-AATGATACGGCGACCACCGAGATCTACACCAGCAGTATGCATGC GCTCGTTTACTATACGAT-3' |
| gDNAWhitelist_Rev | 5'-CAAGCAGAAGACGGCATACGAGAT**NNNNNN**GTGACT GGAGTTCAGACGTGTGCTCTTCCGATCCAAGCGATTCAAAGTTCTATCCG-3' |
| CustomRead1 Primer | 5'-CCAGCAGTATGCATGCGCTCGTTTACTATACGAT-3' |

Reads from the RT-PCR barcode library that contained the WILD-seq barcode motif were identified and the number of unique UMIs supporting each barcode was calculated. Since barcode sequences amplified from gDNA were also available for our 4T1 WILD-seq pool, an additional filtering step was included, and any barcodes not also detected in the gDNA library were excluded from the whitelist. Based on UMI counts, the top 90th percentile of detected barcodes were taken and collapsed for PCR and sequencing errors using hierarchical clustering and combining sequences with a Hamming distance less than 5.

## Single-cell library preparation

Tumour tissues were collected, minced and dissociated using the gentleMACS Octo Dissociator (Miltenyi Biotec) and the relevant kit (Tumor Dissociation Kit mouse). Tissue was processed into single-cell suspensions following manufacturer's instructions and filtered through 70 µm filters (Miltenyi) to remove any remaining larger particles from single-cell suspension after dissociation. The cell suspension was concentrated and filtered again through a 70 µm filter. Three million live cells were sorted based on live-dead staining with propidium iodide to remove dead cells and debris, pelleted and resuspended in 1 mL phosphate-buffered saline with 0.04% bovine serum albumin (Sigma Aldrich). Cells were counted with a hemocytometer to ensure accurate concentration. The final single cell suspension was diluted as required and NGS libraries were prepared using Chromium Single Cell 3' Reagent Kit (v3.1 Chemistry Dual Index, user guide reference: CG000315) with no modifications.

## Enrichment library preparation

To enrich for WILD-seq barcodes, the amplified cDNA libraries were further amplified with WILD-seq-specific primers containing Illumina-compatible adapters and sample indices:

| Name | Sequence |
| --- | --- |
| Enrich_ Fwd | 5'-AATGATACGGCGACCACCGAGATCTACACNNNNNNNNNNNACACTCTTTCCCTACAC GACGCTC-3' |
| Enrich_ Rev | 5'-CAAGCAGAAGACGGCATACGAGATNNNNNNNNNNGTGACTGGAGTTCAGACGTGTGCTCTTC CGATCTCAGCCATGCGCTCGTTTACTATAC-3' |

"N" denotes sample indices

One µL amplified cDNA library was used as template in a 29-cycle PCR reaction using KAPA HiFi HotStart ReadyMix (Roche). To avoid possible PCR-induced library biases, six reactions were run in parallel. All reactions were combined, purified by columns (Gel purification kit, Qiagen) and quantified by Qubit. Gene expression libraries and barcode enrichment libraries were pooled in an approximately 10:1 molar ratio and libraries were sequenced on the NovaSeq platform (Illumina).

## Animals and in vivo dosing

All mouse experiments were performed under the Animals (Scientific Procedures) Act 1986 in accordance with UK Home Office licenses (Project License # PAD85403A) and approved by the Cancer Research UK (CRUK) Cambridge Institute Animal Welfare and Ethical Review Board. Six- to eight-week-old, Female, BALB/c (Strain code 028) or NSG mice (Strain code 614) were purchased from The Charles River Laboratory. Unless otherwise stated, 60,000 tumour cells were resuspended in 50 µL of a 1:1 mixture of PBS and growth-factor reduced Matrigel (Corning). For single cell clones 751 and 1240 250,000 cells were resuspended in 50 µL of a 1:1 mixture of PBS and growth-factor reduced Matrigel (Corning). All orthotopic injections were performed into the fourth mammary gland. Primary tumour

volume was measured using the formula $V=0.5(LxW^2)$, in which $W$ is the width and $L$ is length of the primary tumour.

Tumour-bearing mice were treated with either vehicle or with different drugs from seven days post transplantation. All drugs were administered via intraperitoneal injection. For JQ1 treatment, animals were dosed 75 mg/kg JQ1 (dissolved in DMSO and diluted 1:10 in 10% β-cyclodextrin) 5 days/week (5 consecutive days followed by 2 days off) until tumours reached endpoint. For docetaxel treatment, animals were dosed at 12.5 mg/kg docetaxel (dissolved in 1:1 mixture of ethanol and Kolliphor and diluted 1:4 in saline) 3 times/week for two weeks. L-asparaginase was administered in 100 µL of 60 U L-asparagine (Abcam) diluted in saline. For combination docetaxel/L-asparaginase treatment, D2A1-m2 WILD-seq tumours were treated with 3 doses of 12.5 mg/kg docetaxel (days 7,9,11 post-implantation) and 1 dose of 10 mg/kg docetaxel (day 14 post-implantation). From day 21 mice were treated daily with 60 U per dose of L-asparaginase until end point. For the single clones 751 and 1240, L-asparaginase monotherapy was initiated at day 7 post-implantation and continued daily on a 5 days on 2 days of schedule with each dose consisting of 100 µl of 60 U L-asparaginase or saline vehicle until end-point. Unless otherwise stated vehicle treated mice were sacrificed 21 days post-tumour transplantation and drug treated animals were sacrificed when tumour volumes reached that of vehicle treated animals at 21 days.

## scRNA-seq analysis

scRNA-seq libraries generated by the 10x Chromium platform were processed using CellRanger version 3.0.1. Reads were aligned to a custom reference genome that was created by adding the sequence of the zsGreen-WILD-seq barcode transgene as a new chromosome to the mm10 mouse genome. The gene expression matrices generated were then analyzed with the Seurat R package (*Stuart et al., 2019*) using a standard pipeline. Briefly, datasets were first filtered based on the number of unique genes detected per cell (typical accepted range 200–10000 genes) and the percentage of reads that map to the mitochondrial genome (<12 %). Reads which mapped to the zsGreen-WILD-seq barcode transgene were removed from the count matrix to prevent these driving cell clustering. Normalisation was performed using sctransform, including cell cycle regression. Differential abundance of cell subtypes was performed using Milo (*Dann et al., 2022*).

## Clonal barcode assignment to single cell data

### Extraction of WILD-seq barcodes from scRNA-seq data

Reads mapping to the zsGreen-WILD-seq barcode transgene and containing the full barcode sequence (20nt constant linker with a 12 nt variable region on either side) were extracted from the BAM file produced by Cell Ranger and mapped using Bowtie to a whitelist of barcodes expressed in the WILD-seq cell pool. A WILD-seq clonal barcode was assigned to a cell if there were at least two independent reads which matched the barcode to the cell and more than 50% of barcode mapped reads from the cell supported the assignment.

### Extraction of WILD-seq barcodes from PCR enrichment data

Reads from the PCR barcode enrichment were processed separately using the UMI-tools to extract 10x cell barcodes and UMIs from the raw read files. The sequence corresponding to the full barcode sequence (20nt constant linker with a 12 nt variable region on either side) was extracted from each read and then mapped to the WILD-seq clonal barcode whitelist using Bowtie. A WILD-seq clonal barcode was assigned to a cell if there were at least 10 UMIs which matched the barcode to the cell and at least twice as many UMIs supporting this assignment compared to the next best.

### WILD-seq barcode assignment

The WILD-seq clonal barcode assignment from these two pipelines was then compared. If the assignment from the transcriptomic analysis and the PCR enrichment analysis were in agreement the barcode was assigned. On the rare occasion the assignment didn't match a clonal barcode was not assigned. If a cell was assigned a WILD-seq barcode by only one method, a further more stringent filtering step was included. For WILD-seq barcodes assigned only from the 10x scRNA-seq dataset but not the

PCR-enrichment, the minimum number of UMIs required to support the assignment was increased to 5 and for WILD-seq barcodes assigned only from the PCR-enrichment but not the 10x scRNA-seq dataset, the minimum number of UMIs required to support the assignment was increased to 30.

### Differential gene expression

Differential gene expression was determined using the FindMarkers function in Seurat with a Wilcoxon rank sum test to identify differentially expressed genes. For differential expression of groups of genes, we used the AUCell R package (*Aibar et al., 2017*) which enables analysis of the relative expression of a gene set (i.e. gene signature or pathway) across all the cells in single-cell RNA-seq data using the "Area Under the Curve" (AUC) to calculate the enrichment of the input geneset within the expressed genes for each cell. An AUCell score was calculated for each tumour cell for every gene set in the MSigDB C2 collection (*Liberzon et al., 2011*; *Subramanian et al., 2005*) that contained more than 20 genes with detectable expression in our data. AUCell scores were compared across clones or conditions using a Wilcoxon rank sum test and p-values were adjusted for multiple comparison using the Benjamini-Hochberg correction method.

To generate baseline transcriptomic signatures for each clone in vehicle-treated tumours, comparisons were made between the clone of interest and all assigned tumour cells from the same sample (in the case of D2A1-m2 tumours) or the same experiment (in the case of 4T1 tumours). Samples/experiments were included if they contained at least 20 cells assigned to the clone of interest. To define consistently enriched/depleted signatures, p-values from comparisons within each sample/experiment were combined using the Fisher's method.

### Patient data analysis

Microarray gene expression data was downloaded from GSE28844 (*Vera-Ramirez et al., 2013*). A single probe for each gene was selected based on the highest median expression. Gene set expression per patient sample was calculated using GSVA (*Hänzelmann et al., 2013*).

## Acknowledgements

We thank all members of the Hannon and IMAXT labs at the CRUK Cambridge Institute for valuable discussions and advice, in particular Clare Rebbeck for assistance with home office animal licensing. We also thank CRUK Cambridge Institute's core facilities, in particular members of the genomics core for assistance with sequencing library preparation and sequencing, members of the Biological Resource Unit (BRU) for animal husbandry and members of the flow cytometry core for assistance with cell sorting. GJH is a Royal Society Wolfson Research Professor (RP130039). This study was funded in part by Breast Cancer Now's Catalyst Programme (2021JulyPCC1459) which is supported by funding from Pfizer.

## Additional information

### Group author details

**CRUK IMAXT Grand Challenge Team**
**Gregory J Hannon**: Cancer Research UK Cambridge Institute, Li Ka Shing Centre, University of Cambridge, Cambridge, United Kingdom; **Bruno Albuquerque**: Cancer Research UK Cambridge Institute, Li Ka Shing Centre, University of Cambridge, Cambridge, United Kingdom; **Martina Alini**: Cancer Research UK Cambridge Institute, Li Ka Shing Centre, University of Cambridge, Cambridge, United Kingdom; **Heather Ashmore**: Cancer Research UK Cambridge Institute, Li Ka Shing Centre, University of Cambridge, Cambridge, United Kingdom; **Thomas Ashmore**: Cancer Research UK Cambridge Institute, Li Ka Shing Centre, University of Cambridge, Cambridge, United Kingdom; **Battistoni Giorgia**: Cancer Research UK Cambridge Institute, Li Ka Shing Centre, University of Cambridge, Cambridge, United Kingdom; **Dario Bressan**: Cancer Research UK Cambridge Institute, Li Ka Shing Centre, University of Cambridge, Cambridge, United Kingdom; **Ian Gordon Cannell**: Cancer Research UK Cambridge Institute, Li Ka Shing Centre, University of Cambridge, Cambridge, United Kingdom; **Hannah Casbolt**: Cancer Research UK Cambridge Institute, Li Ka Shing Centre,

University of Cambridge, Cambridge, United Kingdom; **Lauren Deighton**: Cancer Research UK Cambridge Institute, Li Ka Shing Centre, University of Cambridge, Cambridge, United Kingdom; **Ilaria Falciatori**; **Atefeh Fatemi**: Cancer Research UK Cambridge Institute, Li Ka Shing Centre, University of Cambridge, Cambridge, United Kingdom; **Nicole Hemmer**: Cancer Research UK Cambridge Institute, Li Ka Shing Centre, University of Cambridge, Cambridge, United Kingdom; **Cristina Jauset**: Cancer Research UK Cambridge Institute, Li Ka Shing Centre, University of Cambridge, Cambridge, United Kingdom; **Tatjana Kovačević**: Cancer Research UK Cambridge Institute, Li Ka Shing Centre, University of Cambridge, Cambridge, United Kingdom; **Claire M Mulvey**: Cancer Research UK Cambridge Institute, Li Ka Shing Centre, University of Cambridge, Cambridge, United Kingdom; **Natasha Narayanan**; **Fiona Nugent**: Cancer Research UK Cambridge Institute, Li Ka Shing Centre, University of Cambridge, Cambridge, United Kingdom; **Clare Rebbeck**; **Marta Paez Ribes**: Cancer Research UK Cambridge Institute, Li Ka Shing Centre, University of Cambridge, Cambridge, United Kingdom; **Isabella Pearsall**: Cancer Research UK Cambridge Institute, Li Ka Shing Centre, University of Cambridge, Cambridge, United Kingdom; **Sarah Pearsall**; **Fatime Qosaj**: Cancer Research UK Cambridge Institute, Li Ka Shing Centre, University of Cambridge, Cambridge, United Kingdom; **Kirsty Sawicka**: Cancer Research UK Cambridge Institute, Li Ka Shing Centre, University of Cambridge, Cambridge, United Kingdom; **Sophia A Wild**: Cancer Research UK Cambridge Institute, Li Ka Shing Centre, University of Cambridge, Cambridge, United Kingdom; **Elena Williams**; **Hamid Raza Ali**: Cancer Research UK Cambridge Institute, Li Ka Shing Centre, University of Cambridge, Cambridge, United Kingdom; **Samuel Aparicio**: Department of Molecular Oncology, BC Cancer, part of the Provincial Health Services Authority, Vancouver, Canada; Department of Pathology and Laboratory Medicine, University of British Columbia, Vancouver, Canada; **Emma Laks**: Department of Molecular Oncology, BC Cancer, part of the Provincial Health Services Authority, Vancouver, Canada; Department of Pathology and Laboratory Medicine, University of British Columbia, Vancouver, Canada; **Yangguang Li**: Department of Molecular Oncology, BC Cancer, part of the Provincial Health Services Authority, Vancouver, Canada; **Ciara H O'Flanagan**: Department of Molecular Oncology, BC Cancer, part of the Provincial Health Services Authority, Vancouver, Canada; **Austin Smith**: Department of Molecular Oncology, BC Cancer, part of the Provincial Health Services Authority, Vancouver, Canada; **Daniel Lai**: Department of Molecular Oncology, BC Cancer, part of the Provincial Health Services Authority, Vancouver, Canada; Department of Pathology and Laboratory Medicine, University of British Columbia, Vancouver, Canada; **Roth Andrew**: Department of Molecular Oncology, BC Cancer, part of the Provincial Health Services Authority, Vancouver, Canada; Department of Pathology and Laboratory Medicine, University of British Columbia, Vancouver, Canada; Department of Computer Science, University of British Columbia, Vancouver, Canada; **Shankar Balasubramanian**: Cancer Research UK Cambridge Institute, Li Ka Shing Centre, University of Cambridge, Cambridge, United Kingdom; Department of Chemistry, University of Cambridge, Cambridge, United Kingdom; School of Clinical Medicine, University of Cambridge, Cambridge, United Kingdom; **João CF Nogueira**: Cancer Research UK Cambridge Institute, Li Ka Shing Centre, University of Cambridge, Cambridge, United Kingdom; Department of Chemistry, University of Cambridge, Cambridge, United Kingdom; **Max Lee**; **Bernd Bodenmiller**: Cancer Research UK Cambridge Institute, Li Ka Shing Centre, University of Cambridge, Cambridge, United Kingdom; **Marcel Burger**: Cancer Research UK Cambridge Institute, Li Ka Shing Centre, University of Cambridge, Cambridge, United Kingdom; **Laura Kuett**: Cancer Research UK Cambridge Institute, Li Ka Shing Centre, University of Cambridge, Cambridge, United Kingdom; **Jonas Windhager**: Cancer Research UK Cambridge Institute, Li Ka Shing Centre, University of Cambridge, Cambridge, United Kingdom; **Edward S Boyden**: McGovern Institute, Departments of Biological Engineering and Brainand Cognitive Sciences, Massachusetts Institute of Technology, Cambridge, United States; **Debarati Ghosh**: McGovern Institute, Departments of Biological Engineering and Brainand Cognitive Sciences, Massachusetts Institute of Technology, Cambridge, United States; **Anubhav Sinha**: McGovern Institute, Departments of Biological Engineering and Brainand Cognitive Sciences, Massachusetts Institute of Technology, Cambridge, United States; **Brett Pryor**: McGovern Institute, Departments of Biological Engineering and Brainand Cognitive Sciences, Massachusetts Institute of Technology, Cambridge, United States; **Ruihan Zhang**: McGovern Institute, Departments of Biological Engineering and Brainand Cognitive Sciences, Massachusetts Institute of Technology, Cambridge, United States; **Jack Lovell**: McGovern Institute, Departments of Biological Engineering and Brainand Cognitive Sciences, Massachusetts Institute of Technology, Cambridge, United States;

**Chi Zhang**: McGovern Institute, Departments of Biological Engineering and Brainand Cognitive Sciences, Massachusetts Institute of Technology, Cambridge, United States; **Yangning Lu**: McGovern Institute, Departments of Biological Engineering and Brainand Cognitive Sciences, Massachusetts Institute of Technology, Cambridge, United States; **Carlos Caldas**: Department of Oncology and Cancer Research UK Cambridge Institute, University of Cambridge, Cambridge, United Kingdom; **Alejandra Bruna**: Department of Oncology and Cancer Research UK Cambridge Institute, University of Cambridge, Cambridge, United Kingdom; **Maurizio Callari**: Cancer Research UK Cambridge Institute, Li Ka Shing Centre, University of Cambridge, Cambridge, United Kingdom; **Lauren Deighton**: Cancer Research UK Cambridge Institute, Li Ka Shing Centre, University of Cambridge, Cambridge, United Kingdom; **Wendy Greenwood**: Cancer Research UK Cambridge Institute, Li Ka Shing Centre, University of Cambridge, Cambridge, United Kingdom; **Giulia Lerda**: Cancer Research UK Cambridge Institute, Li Ka Shing Centre, University of Cambridge, Cambridge, United Kingdom; **Yaniv Eyal-Lubling**: Department of Oncology and Cancer Research UK Cambridge Institute, University of Cambridge, Cambridge, United Kingdom; **Oscar M Rueda**: Department of Oncology and Cancer Research UK Cambridge Institute, University of Cambridge, Cambridge, United Kingdom; **Abigail Shea**: Department of Oncology and Cancer Research UK Cambridge Institute, University of Cambridge, Cambridge, United Kingdom; **Owen Harris**: Súil Interactive Ltd, Dublin, United Kingdom; **Robby Becker**: Súil Interactive Ltd, Dublin, United Kingdom; **Flaminia Grimaldi**: Súil Interactive Ltd, Dublin, United Kingdom; **Suvi Harris**: Súil Interactive Ltd, Dublin, United Kingdom; **Sara Lisa Vogl**: Súil Interactive Ltd, Dublin, United Kingdom; **Joanna Weselak**: Súil Interactive Ltd, Dublin, United Kingdom; **Johanna A Joyce**: Department of Oncology and Ludwig Institute for Cancer Research, University of Lausanne, Lausanne, Switzerland; **Spencer S Watson**: Department of Oncology and Ludwig Institute for Cancer Research, University of Lausanne, Lausanne, Switzerland; **John Marioni**: Cancer Research UK Cambridge Institute, Li Ka Shing Centre, University of Cambridge, Cambridge, United Kingdom; EMBL-European Bioinformatics Institute, Wellcome Genome Campus, Cambridge, United Kingdom; Wellcome Sanger Institute, Wellcome Genome Campus, Cambridge, United Kingdom; **Sohrab P Shah**: Department of Molecular Oncology, BC Cancer, part of the Provincial Health Services Authority, Vancouver, Canada; Department of Pathology and Laboratory Medicine, University of British Columbia, Vancouver, Canada; Computational Oncology, Department of Epidemiology and Biostatistics, Memorial Sloan Kettering Cancer Center, New York, United States; **Andrew McPherson**: Department of Molecular Oncology, BC Cancer, part of the Provincial Health Services Authority, Vancouver, Canada; Computational Oncology, Department of Epidemiology and Biostatistics, Memorial Sloan Kettering Cancer Center, New York, United States; **Ignacio Vázquez-García**: Computational Oncology, Department of Epidemiology and Biostatistics, Memorial Sloan Kettering Cancer Center, New York, United States; Herbert and Florence Irving Institute for Cancer Dynamics, Columbia University, New York, United States; **Simon Tavaré**: Cancer Research UK Cambridge Institute, Li Ka Shing Centre, University of Cambridge, Cambridge, United Kingdom; Herbert and Florence Irving Institute for Cancer Dynamics, Columbia University, New York, United States; New York Genome Center, New York, United States; **Khanh N Dinh**: Herbert and Florence Irving Institute for Cancer Dynamics, Columbia University, New York, United States; **Russell Kunes**: Herbert and Florence Irving Institute for Cancer Dynamics, Columbia University, New York, United States; **Nicholas A Walton**: Institute of Astronomy, University of Cambridge, Cambridge, United Kingdom; **Mohammad Al Sa'd**: Institute of Astronomy, University of Cambridge, Cambridge, United Kingdom; **Nick Chornay**: Institute of Astronomy, University of Cambridge, Cambridge, United Kingdom; **Ali Dariush**: Institute of Astronomy, University of Cambridge, Cambridge, United Kingdom; **Eduardo A González-Solares**: Institute of Astronomy, University of Cambridge, Cambridge, United Kingdom; **Carlos González-Fernández**: Institute of Astronomy, University of Cambridge, Cambridge, United Kingdom; **Aybüke Küpcü Yoldaş**: Institute of Astronomy, University of Cambridge, Cambridge, United Kingdom; **Alireza Molaeinezhad**: Institute of Astronomy, University of Cambridge, Cambridge, United Kingdom; **Neil Millar**: Institute of Astronomy, University of Cambridge, Cambridge, United Kingdom; **Tristan Whitmarsh**: Institute of Astronomy, University of Cambridge, Cambridge, United Kingdom; **Xiaowei Zhuang**: Howard Hughes Medical Institute, Harvard University, Cambridge, United States; Department of Physics, Harvard University, Cambridge, United States; Department of Chemistry and Chemical Biology, Harvard University, Cambridge, United States; **Jean Fan**: Howard Hughes Medical Institute, Harvard University, Cambridge, United States; Department of Physics, Harvard University, Cambridge, United States; Department of Chemistry and

Chemical Biology, Harvard University, Cambridge, United States; **Hsuan Lee**: Howard Hughes Medical Institute, Harvard University, Cambridge, United States; Department of Physics, Harvard University, Cambridge, United States; Department of Chemistry and Chemical Biology, Harvard University, Cambridge, United States; **Leonardo A Sepúlveda**: Howard Hughes Medical Institute, Harvard University, Cambridge, United States; Department of Physics, Harvard University, Cambridge, United States; Department of Chemistry and Chemical Biology, Harvard University, Cambridge, United States; **Chenglong Xia**: Howard Hughes Medical Institute, Harvard University, Cambridge, United States; Department of Physics, Harvard University, Cambridge, United States; Department of Chemistry and Chemical Biology, Harvard University, Cambridge, United States; **Pu Zheng**: Howard Hughes Medical Institute, Harvard University, Cambridge, United States; Department of Physics, Harvard University, Cambridge, United States; Department of Chemistry and Chemical Biology, Harvard University, Cambridge, United States

## Competing interests

CRUK IMAXT Grand Challenge Team: Gregory J Hannon: is a scientific cofounder, and has equity in, Faeth Therapeutics. The other authors declare that no competing interests exist.

## Funding

| Funder | Grant reference number | Author |
| --- | --- | --- |
| Cancer Research UK | C14303/A21143 C14303/A19926 C14303/A19927 C9545/A24042 | Gregory J Hannon |
| Breast Cancer Now | 2021JulyPCC1459 | Gregory J Hannon |

The funders had no role in study design, data collection and interpretation, or the decision to submit the work for publication.

## Author contributions

Sophia A Wild, Conceptualization, Data curation, Software, Formal analysis, Investigation, Visualization, Methodology, Writing - original draft, Project administration, Writing – review and editing; Ian G Cannell, Conceptualization, Data curation, Formal analysis, Supervision, Investigation, Visualization, Methodology, Writing - original draft, Project administration, Writing – review and editing; Ashley Nicholls, Investigation, Methodology; Katarzyna Kania, Investigation; Dario Bressan, Software, Funding acquisition; CRUK IMAXT Grand Challenge Team, Conceptualization, Funding acquisition; Gregory J Hannon, Conceptualization, Supervision, Funding acquisition, Writing – review and editing; Kirsty Sawicka, Conceptualization, Data curation, Software, Formal analysis, Supervision, Funding acquisition, Investigation, Visualization, Methodology, Writing - original draft, Project administration, Writing – review and editing

## Author ORCIDs

Ian G Cannell ⓘ http://orcid.org/0000-0001-5832-9210
Gregory J Hannon ⓘ http://orcid.org/0000-0003-4021-3898
Kirsty Sawicka ⓘ http://orcid.org/0000-0003-4195-6327
Dario Bressan ⓘ http://orcid.org/0000-0003-3592-699X
Cristina Jauset ⓘ http://orcid.org/0000-0003-1408-026X
Claire M Mulvey ⓘ http://orcid.org/0000-0002-2989-2052
Fiona Nugent ⓘ http://orcid.org/0000-0001-8148-0867
Isabella Pearsall ⓘ http://orcid.org/0000-0001-7771-3082
Sophia A Wild ⓘ http://orcid.org/0000-0003-0397-6255
Hamid Raza Ali ⓘ http://orcid.org/0000-0001-7587-0906
Samuel Aparicio ⓘ http://orcid.org/0000-0002-0487-9599
Ciara H O'Flanagan ⓘ http://orcid.org/0000-0002-4570-7347
Roth Andrew ⓘ http://orcid.org/0000-0003-3422-8823
Marcel Burger ⓘ http://orcid.org/0000-0002-9904-2547
Jonas Windhager ⓘ http://orcid.org/0000-0002-2111-5291
Spencer S Watson ⓘ http://orcid.org/0000-0002-5583-1544
Ignacio Vázquez-García ⓘ http://orcid.org/0000-0003-0427-2639

Simon Tavaré  http://orcid.org/0000-0002-3716-4952
Nicholas A Walton  http://orcid.org/0000-0003-3983-8778
Leonardo A Sepúlveda  http://orcid.org/0000-0003-3602-1009
Chenglong Xia  http://orcid.org/0000-0002-5895-6342

## Ethics

All mouse experiments were performed under the Animals (Scientific Procedures) Act 1986 in accordance with UK Home Office licenses (Project License # PAD85403A) and approved by the Cancer Research UK (CRUK) Cambridge Institute Animal Welfare and Ethical Review Board.

## Decision letter and Author response

Decision letter https://doi.org/10.7554/eLife.80981.sa1
Author response https://doi.org/10.7554/eLife.80981.sa2

---

## Additional files

### Supplementary files

• Supplementary file 1. Overview of single cell RNA-seq samples generated.

• Supplementary file 2. Number and proportion of tumour cells assigned to each clonal barcode for all 4T1 WILD-seq samples.

• Supplementary file 3. Number and proportion of tumour cells assigned to each clonal barcode for all D2A1-m2 WILD-seq samples.

• Supplementary file 4. 4T1 WILD-seq baseline gene enrichment signatures for major clones. Differential gene expression analysis was performed for each clone by comparing cells from a clonal lineage of interest to all assigned tumour cells within the same experiment. Only vehicle-treated samples were included in the analysis. Experiments were included in the analysis if they contained at least 20 cells assigned to the clone and clones were analyzed if they were represented by at least 20 cells in at least 3 of the 4 experiments. Differential gene expression was performed using Seurat FindMarkers function and Wilcoxon Rank Sum test. Fisher's method was used to combine p-values from separate experiments. Analysis for each clone is provided as a separate tab.

• Supplementary file 5. 4T1 WILD-seq baseline gene set enrichment signatures for major clones. Differential gene set expression analysis was performed for each clone by comparing cells from a clonal lineage of interest to all assigned tumour cells within the same experiment. All gene sets from the Molecular Signatures Database C2 curated gene set collection were included in the analysis that contained more than 20 genes detectable in our single cell data. Only vehicle-treated samples were included in the analysis. Experiments were included in the analysis if they contained at least 20 cells assigned to the clone and clones were analyzed if they were represented by at least 20 cells in at least 3 of the 4 experiments. Gene set expression analysis was performed using AUCell and differential expression was calculated using Wilcoxon Rank Sum test. Tables show median AUCell score per experiment for each gene set, enrichment in AUCell score relative to all assigned tumour cells within the same experiment ($\log_2$(median AUCell score clone of interest/median AUCell score all clones)) and adjusted p-value from Wilcoxon Rank Sum test of AUCell scores from clone of interest vs AUCell scores from all assigned tumours cells from the same experiment. Fisher's method was used to combine p-values from separate experiments. Analysis for each clone is provided as a separate tab. A final tab 'Data_for_*Figure 1h'* provides the matrix of AUCell enrichment values used for the heatmap plotted in *Figure 1h* compiled from individual analyses.

• Supplementary file 6. Differential expression analysis JQ1 vs Vehicle. Differential gene expression analysis was performed by comparing cells from the same clonal lineage treated with JQ1 or vehicle within the same experiment. Five clones were included in the analysis (clones 350, 473, 537, 606 and 684) for which there were at least 20 cells per condition across both experiments. Fisher's method was used to combine p-values from different clones within the same experiment. Gene level differential expression was performed using Seurat FindMarkers function and Wilcoxon Rank Sum test. These data are provided under the 'FindMarkers_JQ1vsVeh' tab. Gene set level differential expression was performed using AUCell and differential expression was calculated using Wilcoxon Rank Sum test. These data are provided under the 'AUCell_JQ1vsVeh' tab. The 'Median_norm_AUCell_Scores' tab provides a summary of the median normalised AUCell scores for each clone, condition and experiment used in the preparation of *Figure 2e*. Normalization to enable comparison across separate experiments was performed by dividing by the median AUCell score for all vehicle-

treated tumour cells assigned to any clonal lineage from the same experiment.

• Supplementary file 7. Correlation of clonal gene expression with JQ1 response. To determine genes and gene sets whose expression correlates with JQ1 response, the correlation between baseline gene and geneset enrichment values for the major clones as defined in *Supplementary file 4*, *Supplementary file 5* and the log fold change in clonal abundance between JQ1 and vehicle-treated samples was calculated using the Pearson correlation test. The Pearson correlation coefficient is provided for each gene and gene set.

• Supplementary file 8. Correlation of clonal gene expression with docetaxel response. To determine genes and gene sets whose expression correlates with docetaxel response, the correlation between baseline gene and geneset enrichment values for the major clones as defined in *Supplementary file 4*, *Supplementary file 5* and the log fold change in clonal abundance between JQ1 and vehicle-treated samples was calculated using the Pearson correlation test. The Pearson correlation coefficient is provided for each gene and gene set.

• Supplementary file 9. D2A1-m2 WILD-seq baseline gene enrichment signatures for major clones. Differential gene expression analysis was performed for each clone by comparing cells from a clonal lineage of interest to all assigned tumour cells within the same sample. Only vehicle-treated samples were included in the analysis. Clones were included in the analysis if there were at least 20 cells assigned to that clone in all three vehicle samples (DV1, DV2 and DV3). Differential gene expression was performed using Seurat FindMarkers function and Wilcoxon Rank Sum test. Fisher's method was used to combine p-values from separate samples. Analysis for each clone is provided as a separate tab. In addition, analysis is included for the combined resistant clones 751, 1197 and 1240. Due to their low representation in vehicle-treated samples cells assigned to these clones from all three vehicle-treated samples were combined for gene expression analysis and compared to all assigned tumour cells from the three samples.

• Supplementary file 10. D2A1-m2 WILD-seq baseline gene set enrichment signatures for major clones. Differential gene set expression analysis was performed for each clone by comparing cells from a clonal lineage of interest to all assigned tumour cells within the same sample. All gene sets from the Molecular Signatures Database C2 curated gene set collection were included in the analysis that contained more than 20 genes detectable in our single cell data. Only vehicle-treated samples were included in the analysis. Clones were included in the analysis if there were at least 20 cells assigned to that clone in all three vehicle samples (DV1, DV2 and DV3). Gene set expression analysis was performed using AUCell and differential expression was calculated using Wilcoxon Rank Sum test. Tables show median AUCell score per sample for each gene set, enrichment in AUCell score relative to all assigned tumour cells within the same experiment (log$_2$(median AUCell score clone of interest/median AUCell score all clones)) and adjusted p-value from Wilcoxon Rank Sum test of AUCell scores from clone of interest vs AUCell scores from all assigned tumours cells from the same sample. Fisher's method was used to combine p-values from separate samples. Analysis for each clone is provided as a separate tab. In addition, analysis is included for the combined resistant clones 751 1197 and 1240. Due to their low representation in vehicle-treated samples cells assigned to these clones from all three vehicle-treated samples were combined for gene expression analysis and compared to all assigned tumour cells from the three samples. The tab 'Data for *Figure 4e and f'* provides the matrix of median AUCell scores used for the heatmap plotted in *Figure 4e* compiled from individual analyses. The tab 'Data for *Figure 4h'* provides median AUCell scores per sample for clones of interest for all samples and conditions where at least 20 cells per clone were present. Selected data from this table was plotted in *Figure 4h*.

• Supplementary file 11. Comparison of differential gene expression analysis in bulk tumour cells and intra-clonal changes in gene expression. For each treatment condition (docetaxel/D2A1-m2, docetaxel/4T1 and JQ1/4T1) differential expression analysis was performed between barcoded tumour cells from drug-treated and vehicle-treated animals from the same experiment. Analysis was performed either by using cells from a single clonal lineage (analysis by clone) or all barcoded tumour cells irrespective of clonal lineage (bulk tumour cell analysis). Differential gene expression was performed using Seurat FindMarkers function and Wilcoxon Rank Sum test. Log$_2$ fold change and adjusted p-value are provided for each comparison. For the analysis by clone, the mean logFC of all individual clonal comparisons is given (mean.logFC.clonal) and Fisher's method was used to combine p-values (fisher.combined.pvalue.clonal). Genes were classified as significantly changed in clonal analysis only, bulk analysis only or both analysis methods based on significance cutoffs of p-value <0.05 and logFC <−0.2 or>0.2. Genes identified as significantly changing by one method only met neither logFC nor p-value cutoffs in the alternative method. For analysis of WILD-seq 4T1 data, analysis was performed separately for the 2 experiments and genes had to meet significance cutoffs

in both experiments.

• Supplementary file 12. Overlap of docetaxel resistance markers in 4T1 and D2A1-m2 cell lines. 4T1 resistance genes were defined as those that were significantly enriched in resistant clone 679 but not in sensitive clone 238 (p<0.05). D2A1-m2 resistance genes were defined as those that were significantly enriched in combined resistant clones 1240, 751 and 1197 but not in sensitive clones 118, 2874 or 1072 (p<0.05). Overlap of these lists revealed 47 common genes. These are listed along with their human orthologs.

• Supplementary file 13. Number and proportion of tumour cells assigned to each clonal barcode for docetaxel and L-asparaginase combination experiment.

• Supplementary file 14. Differential expression analysis for L-Asparaginase treatment. Differential gene expression analysis was performed by comparing cells from the same clonal lineage between each DTX +Asp sample and the combined DTX only samples. To ensure there were sufficient cells across all samples, five major clones (118, 1240, 2323, 2874 and 2991) were included in the analysis. Differential expression analysis was performed using Seurat FindMarkers function and Wilcoxon Rank Sum test. Fisher's method was used to combine p-values from different clones within the same comparison. When selecting genes of interest, mean fold change between DTX + ASNase samples and vehicle (also calculated on a per clone basis using abundant clones) was used as an additional cutoff and is included in the table. The most significantly and consistently differentially expressed genes are indicated in the final column 'Meets.cutoffs?'.

• Supplementary file 15. pHSW8 vector sequence. Full annotated sequence file for the WILD-seq library vector pHSW8.

• MDAR checklist

### Data availability

Single cell RNA-seq data have been uploaded to the NCBI Sequence Read Archive (SRA) under BioProject PRJNA797918 and to the Gene Expression Omnibus (GEO) under accession number GSE218774. All analysis used publicly available software and packages as detailed in Materials and Methods. The WILD-seq github repository https://github.com/ksawicka01/WILD-seq (copy archived at swh:1:rev:52ba4d5156b5314ac4d25f9579baf77bbe9a5a77) provides details of code and bioinformatics pipelines used to assign WILD-seq clone barcodes to cells from a typical single cell transcriptomics experiment.

The following datasets were generated:

| Author(s) | Year | Dataset title | Dataset URL | Database and Identifier |
|---|---|---|---|---|
| Wild SA, Hannon GJ | 2022 | PRJNA797918 WILD-seq: Clonal deconvolution of transcriptomic signatures of sensitivity and resistance to cancer therapeutics in vivo | https://www.ncbi.nlm.nih.gov/bioproject/PRJNA797918/ | NCBI BioProject, PRJNA797918 |
| Wild SA, Hannon GJ | 2022 | Clonal transcriptomics identifies mechanisms of chemoresistance and empowers rational design of combination therapies | https://www.ncbi.nlm.nih.gov/geo/query/acc.cgi?acc=GSE218774 | NCBI Gene Expression Omnibus, GSE218774 |

The following previously published dataset was used:

| Author(s) | Year | Dataset title | Dataset URL | Database and Identifier |
|---|---|---|---|---|
| Vera L | 2012 | Differentially expressed genes after treatment with chemotherapy in breast cancer and their correlation with pathologic response | https://www.ncbi.nlm.nih.gov/geo/query/acc.cgi?acc=GSE28844 | NCBI Gene Expression Omnibus, GSE28844 |

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
