## [Editor Report]

This study advances a novel strategy of lineage tracing coupled with single-cell transcriptomics to allow unique insights into tumor heterogeneity and the diversity of response to treatment. These analyses reveal new insights into mechanisms of taxane resistance. Overall, the study is scientifically robust and puts forward a new methodology that will be of interest to scientists as well using this technology to gain insights into the factors that inform resistance to taxane treatment in an in vivo cancer model.

---

## [Decision Letter]

**Decision letter after peer review:**

Thank you for submitting your article "Clonal transcriptomics identifies mechanisms of chemoresistance and empowers rational design of combination therapies" for consideration by *eLife*. Your article has been reviewed by 3 peer reviewers, and the evaluation has been overseen by a Reviewing Editor and W Kimryn Rathmell as the Senior Editor. The following individuals involved in review of your submission have agreed to reveal their identity: Thales Papagiannakopoulos (Reviewer #1); Michael Hemann (Reviewer #2).

All three reviewers were thoroughly impressed with the quality of the manuscript and the potential impact on cancer studies. The authors should be commended for a beautiful piece of work.

Essential revisions:

1) Attend to the requests for clarification and expanded discussion from reviewers 1 and 2, particularly with regard to NRF2 signalling and the EMT/MET transitions.

2) Respond to reviewers requests for additional insights to ATF4 and l-aspariginase sensitivity. Additional analysis of existing data may be warranted, but we are specifically not looking for additional mouse experiments.

3) Please provide a point by point response to the remainder of the reviewer comments.

*Reviewer #1 (Recommendations for the authors):*

– In response to BET inhibitors the authors identify both Unfolded protein response (UPR) and mTOR pathways associated with resistance. UPR involves several signaling arms and transcription factors. The authors should discuss which arm of the UPR pathway is primarily associated with resistance? For example, ATF4 has been previously shown to regulate autophagy which has been shown to lead to resistance to BET inhibition. This is particularly important also in relation to the DTX resistance studies given that ATF4, which is part of UPR, is a well-known regulator of ASNS and amino acid deprivation response.

– The observation that the NRF2 transcriptional signature is upregulated in DTX resistant cells is important. The authors should validate the heterogenous nature of NRF2 activation in tumor pre- and post-taxane DTX treatment either by staining for NRF2 or some of its bona-fide transcriptional targets by immunohistochemistry. They should observe a clonal expansion of the NRF2-high populations upon DTX treatment. Conversely, the authors should observe a selective depletion of these cells in response to L-asparaginase treatment.

– Given that ATF4 is a well known regulator of ASNS, the authors should determine whether ATF4 protein levels are increased in resistant tumors. The authors might not observe an increase of ATF4 transcriptionally given that it is primarily regulated post-transcriptionally. Both ATF4 and ASNS can be detected by immunohistochemistry.

*Reviewer #2 (Recommendations for the authors):*

While I don't know if any of these major experiments are critical for publication (as I think there is substantial, important work presented in the submitted manuscript), I would recommend the following:

Major experiments:

1. Perform transplant experiments with taxane-resistant 4T1 tumors to see if the clonal representation reverts to pre-treatment states.

2. Pretreat D2A1 tumors with L-asparaginase, perform clonal analysis of tumors, and then treat with taxanes to see if pre-treatment eliminates resistant clones (the other possibility is that treatment with taxanes is necessary to create the L-asparaginase sensitive state).

*Reviewer #3 (Recommendations for the authors):*

I think that the experiment is conducted with sufficient rigor, and is exciting in its current state. I have no major recommendations.

---

## [Author Response]

Essential revisions:1) Attend to the requests for clarification and expanded discussion from reviewers 1 and 2, particularly with regard to NRF2 signalling and the EMT/MET transitions.2) Respond to reviewers requests for additional insights to ATF4 and l-aspariginase sensitivity. Additional analysis of existing data may be warranted, but we are specifically not looking for additional mouse experiments.3) Please provide a point by point response to the remainder of the reviewer comments.

We would like to thank the reviewers and editor for their positive and enthusiastic appraisal of our manuscript and we are excited for the opportunity to publish this work in *eLife*. We have carefully considered the reviewers comments and made revisions and additions to the paper as detailed below.

Reviewer #1 (Recommendations for the authors):– In response to BET inhibitors the authors identify both Unfolded protein response (UPR) and mTOR pathways associated with resistance. UPR involves several signaling arms and transcription factors. The authors should discuss which arm of the UPR pathway is primarily associated with resistance? For example, ATF4 has been previously shown to regulate autophagy which has been shown to lead to resistance to BET inhibition. This is particularly important also in relation to the DTX resistance studies given that ATF4, which is part of UPR, is a well-known regulator of ASNS and amino acid deprivation response.

In the context of clonal resistance to JQ1 resistance, the correlation with UPR gene expression appears to be driven by the IRE1 branch of the UPR and not the ATF4/PERK or ATF6 pathways (see new Figure 2 —figure supplement 1). In the context of L-asparaginase resistance, as suggested by the reviewer, ATF4 activation most likely underlies the upregulation of *Asns* in response to amino acid deprivation. In tumour cells treated with asparaginase we see strong correlation between ATF4 activity (as assessed by expression levels of transcriptional targets of ATF4) and *Asns* expression (See new Figure 7f and Figure 7 —figure supplement 1b).

– The observation that the NRF2 transcriptional signature is upregulated in DTX resistant cells is important. The authors should validate the heterogenous nature of NRF2 activation in tumor pre- and post-taxane DTX treatment either by staining for NRF2 or some of its bona-fide transcriptional targets by immunohistochemistry. They should observe a clonal expansion of the NRF2-high populations upon DTX treatment. Conversely, the authors should observe a selective depletion of these cells in response to L-asparaginase treatment.

We agree with the reviewer that this would be a nice secondary validation of the NRF2-high nature of these clones and their expansion with DTX treatment, however, unfortunately, we dissociated the whole tumour when we collect samples for scRNA-Seq to avoid any regional/spatial bias in sampling of the cells. As such, we don’t have samples available that would be suitable for immunohistochemistry.

– Given that ATF4 is a well known regulator of ASNS, the authors should determine whether ATF4 protein levels are increased in resistant tumors. The authors might not observe an increase of ATF4 transcriptionally given that it is primarily regulated post-transcriptionally. Both ATF4 and ASNS can be detected by immunohistochemistry.

Similar to above, we do not have samples available for the proposed immunohistochemistry experiment. Instead, we have analysed the output of the ATF4 pathway in our scRNA-Seq data. We find that expression of transcriptional targets of ATF4 correlate very well with *Asns* specifically under asparaginase treatment (See new Figure 7f and Figure 7 —figure supplement 1b) suggesting a role for ATF4 in mediating the *Asns* up-regulation in this setting.

Reviewer #2 (Recommendations for the authors):While I don't know if any of these major experiments are critical for publication (as I think there is substantial, important work presented in the submitted manuscript), I would recommend the following:Major experiments:1. Perform transplant experiments with taxane-resistant 4T1 tumors to see if the clonal representation reverts to pre-treatment states.

We agree this would be an interesting avenue for future work but is beyond the scope of this report.

2. Pretreat D2A1 tumors with L-asparaginase, perform clonal analysis of tumors, and then treat with taxanes to see if pre-treatment eliminates resistant clones (the other possibility is that treatment with taxanes is necessary to create the L-asparaginase sensitive state).

We have added new data that show NRF2-high, taxane-resistant clones are sensitive to L-asparaginase in the absence of taxane treatment. See response to point 2 above.